



# Coupled large eddy simulations of land surface heterogeneity effects and diurnal evolution of late summer and early autumn atmospheric boundary layers during the CHEESEHEAD19 field campaign

Sreenath Paleri[1,2,3], Luise Wanner[4,5], Matthias Sühring[6,7], Ankur Desai[1], and Matthias Mauder[4,5]

[1]Department of Atmospheric and Oceanic Sciences, University of Wisconsin-Madison, Madison, Wisconsin, USA
[2]Now at Cooperative Institute for Severe and High-Impact Weather Research and Operations, University of Oklahoma, Norman, Oklahoma, USA
[3]Now at Atmospheric Turbulence and Diffusion Division, NOAA/Air Resources Laboratory, Oak Ridge, Tennessee, USA
[4]Institute of Hydrology and Meteorology, Technische Universität Dresden, Dresden, Germany
[5]Institute of Meteorology and Climate Research – Atmospheric Environmental Research, Karlsruhe Institute of Technology, Garmisch-Partenkirchen, Germany
[6]Institute of Meteorology and Climatology, Leibniz University Hannover, Hannover, Germany
[7]Pecanode GmbH, Goslar, Germany

**Correspondence:** Sreenath Paleri (paleri@wisc.edu)

**Abstract.** Observational studies and large eddy simulations (LES) have reported secondary circulations in the turbulent atmospheric boundary layer (ABL). These circulations form as coherent turbulent structures or mesoscale circulations induced by gradients of land surface properties. However, simulations have been limited in their ability to represent these events and their diurnal evolution over realistic and heterogeneous land surfaces. In this study, we present a LES framework combining

the high-resolution observational data collected as part of the CHEESEHEAD19 field campaign to overcome this gap and test how heterogeneity influences the ABL response. We simulated diurnal cycles for four days chosen from late summer to early autumn over a large ($49\times52$ km) heterogeneous domain. To investigate surface atmospheric feedbacks such as self-reinforcement of mesoscale circulations over the heterogeneous domain, the simulations were forced with an interactive land surface model with coupled soil, radiative transfer and plant canopy model. The lateral and model top boundary conditions

were prepared from National Oceanic and Atmospheric Administration High-Resolution Rapid Refresh (HRRR) meteorological analysis fields. Comparing the simulated profile and near surface data with field measured radiosonde and eddy covariance station data showed a realistic evolution of the near-surface meteorological fields, heat and moisture fluxes and the ABL. The LES had limitations in simulating the night-time cooling in the nocturnal boundary layer. The simulated fields were strongly modulated by the imposed HRRR derived mesoscale boundary conditions, resulting in a slightly warmer and drier ABL. The

simulations were run without clouds which resulted in higher daytime sensible heat fluxes for some scenarios.

Our findings demonstrate the capability of the PALM model system to realistically represent the daytime evolution of the ABL response over unstructured heterogeneity and the limitations involved therein with respect to the role of boundary conditions and the representation of the nocturnal boundary layer. The simulation setup and dataset described in this manuscript build the baseline to tackle specific research questions associated with the CHEESEHEAD19 campaign, particularly to address ques-

tions about the role of heterogeneous ecosystems in modulating surface-atmosphere fluxes and near surface meteorological



fields as well as highlight the needed improvements in model representations of land-atmosphere feedbacks over vegetated environments.

## 1 Introduction

Heterogeneity is pervasive over land surfaces, whose properties can hence vary at a wide range of spatio-temporal scales (Paleri et al., 2022; Desai et al., 2022b; Paleri et al., 2023a). This phenomenon means that atmospheric boundary layer (ABL) flow can be perturbed not just at one specific length or time scale but at multiple scales, which excites physical processes such as advection, secondary circulations, and non-equilibrium turbulence (Bou-Zeid et al., 2020). Further, the issue of land surface heterogeneity is fundamentally an issue of scale (Mahrt, 2000; Bou-Zeid et al., 2020) coupled to local biogeophysical processes, as well as larger scale synoptic circulations. Observing and quantifying these interactions and their role in ABL development calls for space and scale intensive deployments seldom seen in micrometeorological field experiments (Wulfmeyer et al., 2018). Observational studies and large eddy simulations (LES) to understand the interactions between landscape heterogeneities and secondary circulations in the ABL have been limited in their ability to capture these events and robustly test hypotheses.

There have not been a lot of modelling studies exploring the surface-atmospheric coupling in a diurnally evolving boundary layer over realistic, irregular heterogeneities. Huang and Margulis (2010) performed a 12 hour daytime LES of the Convective Boundary Layer (CBL) for the Soil Moisture–Atmosphere Coupling Experiment 2002 and evaluated the simulations against experimental measurements. Their results showed up to 18% errors in surface sensible heat fluxes when surface-atmospheric coupling is ignored. In their LES investigation of the diurnal ABL for the Horizontal Array Turbulence Study (HATS) field experiment using prescribed and homogeneous surface fluxes Kumar et al. (2006) found that local scaling approaches were appropriate to describe the free convective and transitioning ABL. LeMone et al. (2010a, b) performed simulations of CBL conditions from the 2002 International H2O Project (IHOP 2002) using the advanced research version of the Weather Research and Forecasting model (WRF-ARW) coupled to the Noah land surface model. Their simulations were able to capture mesoscale circulations during clear sky days ($\sim$ 1 - 10 km) consistent with aircraft measurements. Maronga and Raasch (2013) (hereafter referred to as MR13) investigated the role of irregular and multi scale surface heterogeneities on the ABL using LES of the Lindenberg Inhomogeneous Terrain Fluxes between Atmosphere and Surface - 2003 field experiment days using the PArallelized LES Model (PALM). They used prescribed surface fluxes, prepared from eddy covariance tower measurements and land surface class distributions within their 20 km × 20 km study area with cyclic boundary conditions and a range of imposed geostrophic winds (2, 3, 4, 6 m s$^{-1}$). Using time-ensemble averaging of their 3D LES data, they diagnose signals of surface heterogeneity induced secondary circulations, that were roll-like and aligned along the mean wind for strong mean winds and localised for CBL with weaker horizontal winds.

Such coherent secondary circulations, super imposed over the primary atmospheric turbulence, have been reported in observations (Lemone, 1973, 1976; Weckwerth et al., 1997; Eder et al., 2015) and numerical simulations (Avissar and Schmidt, 1998; Khanna and Brasseur, 1998). Even over homogeneous surface conditions, they can organise as horizontal convective rolls (HCR) (Lemone, 1973; Etling and Brown, 1993) in forced convective boundary layers with strong mean wind-shear and



into convective cells alike turbulent Rayleigh-Benard convective cells (Atkinson and Wu Zhang, 1996; Salesky et al., 2017) in
free convective boundary layers, when the mean wind shear is weaker and surface fluxes stronger.

However, prior research focus has largely been on simulating the ABL response over simplistic or idealised surface het-
erogeneities and forced by cyclic boundary conditions. The presence of land surface gradients in heterogeneous surfaces
can trigger thermally induced mesoscale circulations (TMC) that can be quasi-localised over these land surface temperature
boundaries (Blanford et al., 1991; Foken, 2008; Maronga and Raasch, 2013; Kenny et al., 2017) and slowly move with the
imposed mean wind (Charuchittipan et al., 2014). In their simulations of the LITFASS field experiments, MR13 showed that
the transport due to these circulations can be important during free convective conditions, with the heterogeneity induced heat
flux maximum reaching up to 20% of total surface fluxes and the heterogeneity induced moisture flux maximum reaching up
to 39% of total surface fluxes. Under such conditions the ABL turbulence is not homogeneous, complicating applications of
Taylor's frozen turbulence and Monin-Obukhov Similarity Theory (MOST) for measurement and modelling of ABL processes.
The PALM model system (Maronga et al., 2020a) has since been widely used to study the relationship between land surface
heterogeneities and secondary circulations in the ABL (Kanani-Sühring and Raasch, 2017; Gronemeier et al., 2017; Kröniger
et al., 2018; De Roo and Mauder, 2018; Xu et al., 2020; Akinlabi et al., 2022).

The Chequamegon Heterogeneous Ecosystem Energy-balance Study Enabled by a High-density Extensive Array of Detec-
tors 2019 (CHEESEHEAD19) was a field campaign designed to intensively sample and scale land surface properties and the
ABL response to it over a core $10{\times}10$ km$^2$ heterogeneous domain in Northern Wisconsin (USA) from June until October 2019
(Butterworth et al., 2021; Metzger et al., 2021). The field campaign had three Intensive Observation Periods (IOPs) of one week
each, spread across three months, to intensively sample the spatial heterogeneity and seasonal shift as the domain transitions
from late summer to early autumn. Spatially resolved airborne eddy covariance measurements revealed persistent contributions
of larger, landscape scale (in the range of meso-$\beta$ to meso-$\gamma$ ) fluxes to the daytime sensible and latent heat fluxes (Paleri et al.,
2022). We have a comprehensive dataset at hand that allows us to investigate the land-atmospheric coupling through surface
heterogeneity induced secondary circulations. Due to the large range of scales, diurnal nature of surface sources and three-
dimensionality of the problem, realistic turbulence-resolving numerical simulations of field experiment days can help to gain
further insights to the underlying spatio-temporal processes as the system can be studied as a whole. Unlike prior studies, we
aim to simulate the observed ABL diurnal evolution in a LES with an interactive land surface model (LSM) coupled to plant
canopy and using non cyclical boundary conditions, representative of field experiment days over the heterogeneous forested
landscape. The simulations can also be compared with the intensive observational data the high density eddy covariance station
network characterising the near surface atmospheric state.

As a first step towards understanding the diurnal evolution of surface heterogeneity induced coherent structures in the ABL,
we present and evaluate the CHEESEHEAD19 LES framework. We use information about real world surface heterogeneity
over the predominantly forested study domain, with multiple scales of variability in surface properties, and mesoscale model
forcings during the field experiment, to simulate diurnal cycles of two days during the IOPs of the field campaign. The days
chosen are two days in late summer and two days in autumn. This design helps us to capture the shift in the surface energy
budget partitioning of the study domain as it shifts from a latent heat flux dominated, free convective boundary layer in the



summer to a forced convective boundary layer in autumn with lower Bowen ratios (as reported from aircraft measurements in

Paleri et al. (2022) and tower measurements in Butterworth et al. (2021)).

We hypothesise that LES, initialised with realistic surface heterogeneity, develop similar mesoscale structures and patterns as observed in reality. Following through, we ask, can such a LES be used to evaluate mechanisms that generate surface-heterogeneity induced mesoscale circulations in the diurnal ABL ? This manuscript is organised as follows: Section 2 briefly describes the field measurements used to inform the LES setup. Section 3 goes into the description of the LES modules setup.

The simulation results are compared with field measurements in Sect. 4 and discussed in Sect. 5. A summary and outlook for future studies are presented in Sect. 7.

## 2   Field Experiment Data

### 2.1   Radiosonde Measurements

During the IOPs, atmospheric profiling was performed by the National Center for Atmospheric Research (NCAR) Earth Ob-

serving Laboratory (EOL) Integrated Sounding System (ISS) and the University of Wisconsin – Madison Space Science and Engineering Center Portable Atmospheric Research Center (SPARC) using the Vaisala RS41-SGP radiosondes. The data was processed by the Vaisala MW41 sounding system and the quality controlled data is hosted at the NCAR EOL CHEESE-HEAD19 repository (Facility and , SSEC) During the IOP days there were 4 radisonde launches per day. On 23 August these were at 0610, 0915, 1300 and 1645 CDT. On 24 September the launches were at 0615, 0913, 1300 and 1645 CDT. The

radiosonde profile data were linearly interpolated to the Child01 model vertical grid for inter-comparisons.

### 2.2   Tower and Plot-Level Measurements

The EC tower data used in this study was collected from 17 flux tower sites, set up as part of the National Center for Atmospheric Research (NCAR)-Integrated Surface Flux Station (ISFS) network, and the AmeriFlux US-PFa site within the core 10 km × 10 km CHEESEHEAD19 study domain (Fig. 3b and Table ES2 in Butterworth et al. (2021)). Ground based

measurements of forest canopy leaf phenology were done at 51 plots within the domain (Schwartz, M.D et al. 2019). The tower network collected measurements of 3D-wind, temperature, moisture and $CO_2$ at 20 Hz using the Campbell Scientific CSAT3AW sonic anemometer and the open path Campbell Scientific EC150 infra-red gas analyser. The processed and quality controlled 30-minute flux data are hosted in the AmeriFlux repository. Drone-based LiDAR measurements were performed at 9 select forested flux tower sites to characterise the three-dimensional forest structure (Table 1 in Murphy et al. (2022)). The

measurements were made from 25-29 June 2019, at 60 m above ground using a Routescene discrete-return LiDAR onboard a UAS hexacopter DJI M600 Pro to collect high-density 3D scans ($\sim 600$ points $\mathrm{m}^{-2}$).



## 3 Large Eddy Simulations Model Setup

We used the PArallelized LES Model (PALM) V 6.0 (Maronga et al., 2020a) revision number 21.10-rc.2 for the numerical simulations. PALM solves the non-hydrostatic, filtered, incompressible Navier-Stokes equations in Boussinesq-approximated form on an Arakawa staggered C-grid (Harlow and Welch, 1965; Arakawa and Lamb, 1977). Prognostic equations are solved for the velocity components $u, v, w$, the potential temperature $\theta$, water vapour mixing ratio $q$, a passive scalar $s$ and the subgrid-scale turbulent kinetic energy (SGS-TKE) $e$ . The sub-grid scale terms are parameterised using the 1.5 order turbulent kinetic energy scheme of Deardorff (1980) modified by Moeng and Wyngaard (1988) and Saiki et al. (2000). The advection terms were discretised using a fifth-order scheme (Wicker and Skamarock, 2002), and a third-order Runge–Kutta scheme by Williamson (1980) was used for the time integration.

The dynamical core of the model system and additional modules that enable modelling capabilities for diel surface-atmospheric exchanges over realistic surface have been extensively validated (Heinze et al., 2017; Resler et al., 2021; Gronemeier et al., 2021). Resler et al. (2021) presents a validation of PALM simulations, constrained by field measurements and boundary conditions driven by WRF outputs of the diel boundary layer over Prague, Czech Republic. Wanner et al. (2021) did a comparative simulation study using PALM, with varying surface boundary conditions that included prescribed surface fluxes, a coupled Land Surface Model (LSM) and a coupled LSM combined with a Plant Canopy Model (PCM). Based on their idealised simulation results, they recommend the used of a LSM combined with a PCM as the surface boundary condition to better represent the near-surface dispersive fluxes associated with secondary circulations.

Simulations of the diurnal cycle for two consecutive CHEESEHEAD19 IOP days during the summer and autumn IOPs were performed. August 22 and 23 were chosen for the August IOP and September 24 and 25 for the September IOP. During the August IOP, towards the end of summer, the CHEESEHEAD19 experimental conditions were more free convective than during the September IOP, when they were more forced convective (Paleri et al., 2022). So choosing these two test cases lets us compare and contrast the secondary circulations between two different dynamic stability regimes. Eight ensemble member runs were performed for each of the IOP test cases. At the start of each ensemble member model run, the imposed perturbations by the synthetic turbulence generator (Sect. 3.1) are randomly generated, which in turns produces a unique turbulence realisation for each ensemble member. The Bulk Cloud Physics model within the PALM model system was not switched on for these runs. Domain mean surface incoming shortwave and longwave radiation values were prescribed which were passed to PALM's Radiative Transfer Model (Sect. 3.3.3) to model the influences of the heterogeneous plant canopy and high clouds on surface radiative fluxes (Krč et al., 2021). The maximum model timestep was kept fixed at 0.6 s. The simulations were performed in the NCAR Cheyenne supercomputer (Computational and Laboratory, 2017) and the Casper data analysis and visualisation cluster was used for 3D visualisations and exploratory data analysis. A total of 6,500,000 core hours were used for the production runs.



## 3.1 Large Scale Forcing, Boundary Conditions:

In order to simulate the diurnal variations during the field experiment, with evolving synoptic conditions over a heterogeneous
study domain, non-stationary lateral and top boundary Dirichlet conditions were prescribed for the simulations using an offline-
nesting approach (Kadasch et al., 2021). The large scale forcing data for the boundary conditions were prepared using the
National Centers for Environmental Prediction - High Resolution Rapid Refresh (NCEP-HRRRv4) version of the Weather
Research and Forecasting - Advanced Research WRF (WRF-ARW) model (Benjamin et al., 2016) with a native horizontal
resolution of 3 km. 2D surface and 3D isobaric data were extracted from the University of Utah HRRR archive (Blaylock et al.,
2017). The HRRR data was extracted over the $49 \times 50$ km$^2$ Parent simulation domain (Sect. 3.2) and averaged horizontally to
1D profile data. The isobaric HRRR data were interpolated to the PALM model grid by adapting the open-source WRF4PALM
v1.0 code (Lin et al., 2021). These profiles are prescribed at the model lateral and top boundaries as non-cyclic conditions,
refreshed every hour. The forcing-profiles are read and mapped onto the boundaries during the simulation as well as interpolated
linearly in time. A Synthetic Turbulence Generator (STG) module was used to impose turbulent fluctuations at the model
boundaries in conjunction with the mesoscale forcings since the imposed boundary conditions do not have information about
boundary layer turbulence. Kadasch et al. (2021) give a detailed description of PALM's internal routines which read and process
the initial and boundary conditions as well as the STG. Turbulence is triggered at the model lateral boundaries by imposing
spatial and time varying perturbations to the velocity components with characteristic amplitude depending on the Reynolds
stress tensor following Xie and Castro (2008), as modified by Kim et al. (2013) and the Reynolds stress parameterisation of
Rotach et al. (1996).

With regard to surface forcings, the initial gridded surface soil temperature and moisture values were extracted from the
HRRR data and regridded to match the Parent domain horizontal grid. PALM can support nested LES domains that are re-
cursively nested within and run parallel to each other with continuous communication at run time through its self-nesting
capabilities (Hellsten et al., 2021). Self-nesting was applied in combination with the offline-nesting, with the outermost Par-
ent model receiving input from the larger-scale model and two children (Child01 and Child02, Fig. 1, Sect. 3.2) recursively
nested within the outermost Parent model (Fig. 1). In this setup, only the outermost Parent model requires the initialisation and
boundary data from the larger-scale model. One-way nesting was employed between the three models, where only the parent
communicates with its immediate child at the model lateral and top boundaries at every time step and child model solutions do
not affect flow in its Parent model. Employing both the offline-nesting and self-nesting modules lets us include the synoptic-
scale effects over the simulation domain and model the influence of a heterogeneous land surface and plant canopy over a wide
range of scales.

## 3.2 Domain Resolutions And Nesting Setup

A rather coarse Parent domain covering a $48.6 \times 52$ km$^2$ region, centred around the Department Of Energy Ameriflux regional
tall tower (US-PFa $45.9459°$ N, $-90.2723°$ W, Desai et al. (2022a)) was set up with a horizontal resolution of 90 m to guarantee
that turbulence is fully developed in the area of interest. Since the large-scale forcing data does not have resolved turbulence,



adjustment zones are needed to allow for turbulence development at the inflow boundary so that realistic turbulence is simulated within the domain of interest. Initial test simulations were done for just the Parent domain and the $e$ horizontal profiles from these simulations show that turbulent flow is well developed after about 10 km downstream of the inflow boundary (Fig.A1). At 11:00 hours simulation time, the boundary layer is still in a transition phase from stable to convective conditions so that the

horizontal $e$ profile shows variations, but at 13:00 hours and 15:00 hours it clearly showed an equilibrium value. Hence, the Child01 domain was set up covering a $27 \times 32$ km$^2$ region inside the parent domain, ensuring that the domain is sufficiently far away from the inflow boundaries. This Child01 domain also covered the greater-CHEESEHEAD19 study domain where airborne turbulence measurements were conducted during the IOPs. Inside the Child01 domain, a fine resolution Child02 model was set up, with $12 \times 12$ km$^2$ horizontal extent, covering the core CHEESEHEAD19 experiment study area (Fig. 1).

Simulations for the August IOP days were started at 00:00 Central Daylight Time (CDT = UTC - 6h, abbreviated as CDT from now) 22 August 2019 and went on for 44 hours till 20:00 CDT on 23 August 2019. Likewise, simulations for the September IOP days were started at 00:00 23 September 2019 CDT and went on for 44 hours till 20:00 CDT on 24 September 2019. Table 1 gives an overview of the model resolutions used for the August and September IOP simulations. The horizontal extents were kept the same for both the IOP test cases and all 3 nested models. Parent model vertical extents were changed

between the two IOP cases reflecting the change in the maximum boundary layer heights between the summer and autumn IOP while optimising the available computational resources and ensuring that all of the ABL was included in the simulations. For the August IOP runs, the Parent model vertical extent was fixed at 5000 m, with vertical grid stretching starting at 3000 m to optimise computational resources, with the maximum vertical grid spacing set to 60 m. The Parent model domain for the September IOP runs had a vertical domain extent of 3050 m, with grid stretching starting at 1800 m and the maximum vertical

grid spacing set again to 60 m. In the Child01 model domain, the vertical model extent was set to ensure that at least one of the IOP days simulated would have a fully resolved CBL. The Child02 model domain has a vertical extent of 240 m for both the IOP test cases. Its horizontal resolution of 6 m and a vertical resolution of 4 m allowed us to have a resolved plant canopy in this finer resolution domain.

In this manuscript we focus on 3D data from the Child01 model for 23 August and 24 September simulations, when the model

domain encompassed the whole of CBL (Sect. 4.1).



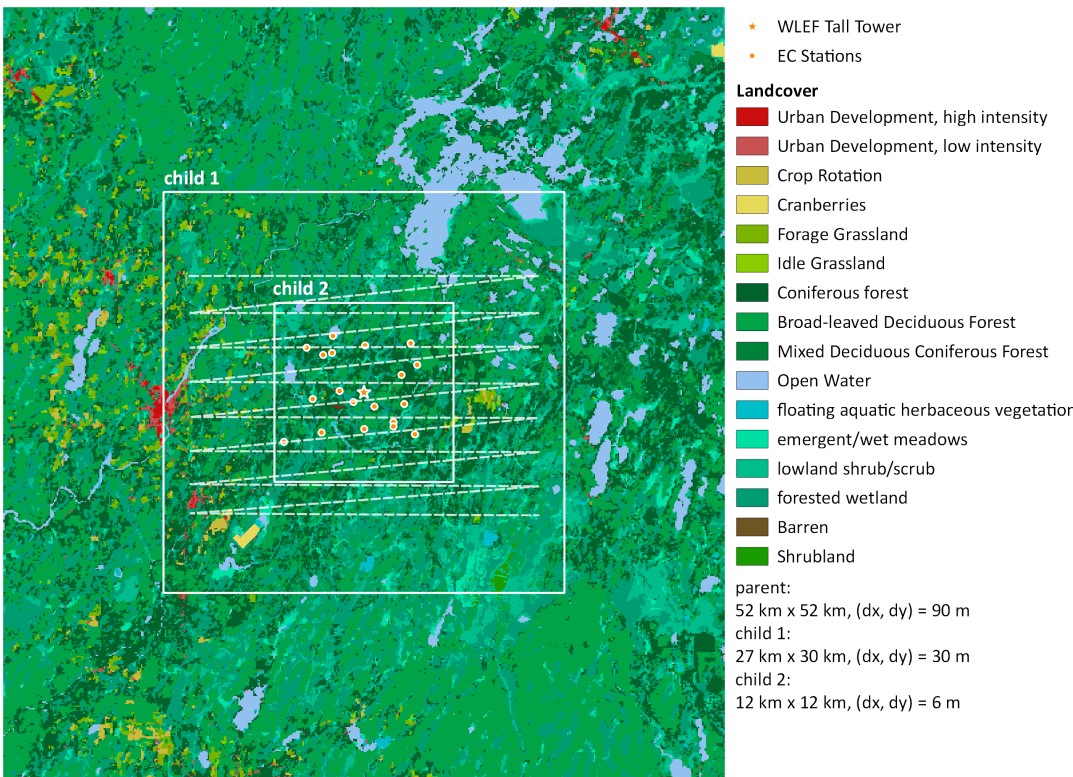

**Figure 1.** Land surface class distribution within the Parent model domain. Child01 model is shown as a $27 \times 30$ km$^2$ box outlined in white. Child02 model domain is shown as a white outlined $12 \times 12$ km$^2$ box within Child01. The orange dots within Child02 denote the CHEESEHEAD19 tower locations. The orange star indicates the location of the US-PFa tall tower. The dotted white line over Child02 is a representative virtual flight pattern which was set up in Child01. Map credit Luise Wanner.





**Table 1.** Grid resolution and domain extents for both August and September IOP test cases. lx, ly, lz denote the spatial extents along a Cartesian coordinate system centred at the south west corner of each model domain.

| Simulation time (CDT) | | Domain | Horizontal extent (km) | Vertical extent (km) | Grid spacing (m) | Grid points |
|---|---|---|---|---|---|---|
| Start | End | | | | | |
| 08.22\|00:00 | 08.23\|20:00 | Parent | lx = 48.6, ly = 51.84 | lz = 5.0 | dx = dy = 90, dz = 12 | nx = 540, ny = 576, nz = 296 |
| | | Child01 | lx = 27, ly = 30.240 | lz =2.49 | dx = dy = 30, dz = 12 | nx = 900, ny = 1008, nz = 248 |
| | | Child02 | lx = ly =12 | lz = 0.24 | dx = dy = 6, dy = 4 | nx = ny = 2000, nz = 60 |
| 09.24\|00:00 | 09.25\|20:00 | Parent | lx = 48.6, ly = 51.84 | lz = 3.05 | dx = dy = 90, dz = 12 | nx = 540, ny = 576, nz = 180 |
| | | Child01 | lx = 27, ly = 30.240 | lz =1.8 | dx = dy = 30, dz = 12 | nx = 900, ny = 1008, nz = 148 |
| | | Child02 | lx = ly =12 | lz = 0.24 | dx = dy = 6, dy = 4 | nx = ny = 2000, nz = 60 |

## 3.3 Land Surface and Plant Canopy Setup

### 3.3.1 Land Surface Model

To accurately simulate the physical processes as observed during the IOPs as realistic as possible, we apply a Land Surface Model (LSM) with coupled soil, radiative transfer and a Plant Canopy Model (PCM). The use of the LSM and PCM runs instead of prescribed surface fluxes enables the investigation of surface atmospheric feedback such as self-reinforcement of mesoscale circulations over the heterogeneous study domain (Wanner et al., 2021). This offers a direct interaction with the synoptic and radiative forcing.

The inbuilt LSM implementation is based on the European Centre for Medium-Range Weather Forecasts-Integrated Forecast System (ECMWF-IFS) land surface parameterisation (H-TESSEL) and its adaptation in the Dutch Atmospheric LES model (Heus et al., 2010). Gehrke et al. (2021) presents a detailed description of the LSM implementation in PALM (Their Tables 1, 2 and 3 give a detailed description of PALM's land surface classes) and presents evaluations against in-situ measurements. The soil type for the entire domain was kept a constant. We used a 'medium fine' texture following the ECMWF-IFS classification that is based on the parameterisation after Van Genuchten (1980). The soil model was configured as an eight layer model. Having a thin soil layer can lead to unrealistic high-frequency feedback effects with near surface atmospheric temperature and moisture (Liu and Shao, 2013). Each layer thickness in metres starting from top was set as: 0.01, 0.02, 0.04, 0.06, 0.14, 0.26, 0.54 and 1.86. The 2D initial and deep soil temperatures and initial soil moisture was taken from the HRRR data (Sect. 2.1). The distribution of different land cover types within the domain is based on the Wiscland 2.0 digital database (Wisconsin Department of Natural Resources, Fig. 1) which is a land cover map of the entire state of Wisconsin with a resolution of 30 m





and providing multiple levels of detail. The data was collected from 2014 to 2016. Wiscland 2.0 landcover types were grouped
into 18 categories (Table 2)

Before the simulations start we used a 44 hour spin up for the LSM where only the surface-soil model is updated (based on
the HRRR data), while the wind speed is held constant at its initial time value and the temperature experiences the daily cycle.
This allows for the near surface soil moisture and temperature values and atmospheric values to come to an equilibrium, which
would be important over the heterogeneous domain as shown by Maronga et al. (2020b).

**Table 2.** Wiscland land surface classes and their assigned PALM LSM classes

| Wiscland Land surface type | PALM Surface Type |
|---|---|
| Open Water | lake |
| Floating Aquatic Herbaceous Vegetation | lake |
| Developed, High intensity | concrete |
| Developed, Low intensity | asphalt/concrete mix |
| Barren | bare soil |
| Crop Rotation | crops, mixed farming |
| Cranberries | irrigated crops |
| Grassland | tall grass |
| Emergent/Wet Meadow | tall grass |
| Shrubland | deciduous shrubs |
| Lowland Scrub/Shrub | deciduous shrubs |
| Coniferous Forest | short grass[†] |
| Coniferous Forested Wetland | short grass[†] |
| Broad-leaved Deciduous Forest | short grass[†] |
| Forested Wetland - Deciduous Forest | short grass[†] |
| Mixed Deciduous/Coniferous Forest | short grass[†] |
| Mixed Deciduous/Coniferous Forested Wetland | short grass[†] |

[†]To avoid double-accounting for surface radiative effects of the canopy, the forested domains are assigned 'short-grass' classes in the LSM setup

### 3.3.2  Plant Canopy Model

The plant canopy is modelled as a porous viscous medium that removes momentum from the flow (Shaw and Schumann, 1992;
Watanabe, 2004), and acts as source/sink for heat, humidity, or passive scalar. We use the land-surface model at the surface of
the entire domain and stack the plant canopy model on top in areas covered by trees. [†]To avoid double-accounting for surface
radiative effects of the canopy, the forested domains are assigned 'short-grass' classes in the LSM setup. The forested areas are
vertically resolved by up to 12 grid layers in the Child02 domain and up to 5 grid layers in the Parent and Child01 domains.
The leaf-area density (LAD) profiles are based on field observations at 9 forested sites within the CHEESEHEAD19 study
area (Sect. 2.2). Here, LAD denotes the area of plant canopy in a given grid cell, $m^2$ / $m^3$. We grouped these observations
into broadleaf, needle leaf and mixed forests. Within each group, the LAD profiles were averaged and extrapolated toward the



ground since no measurements were available near the surface. These profiles are shown in Fig. 2 a, b, c. The profiles were
then scaled in height so that they correspond to the mean canopy heights that were found near the eddy covariance stations
within the individual forest types and finally re-sampled to the PALM grid spacing. The resulting LAD profiles are shown in
Fig. 3.

**Figure 2.** Drone measured LAD profiles classified as a) broadleaf, b) needle leaf, and c) mixed forests. Profiles were linearly extrapolated
from the lowest measurement value towards the surface. Measured leaf fall percentage for a standard (e) and wetland (e) forest in the study
domain. Different colours indicate individual measurements and the black line indicates the mean across trees.

To adapt the LAD profiles to seasonal changes in leaf-area index (LAI), we used phenological observations of leaf fall that
were also carried out at different plots within the CHEESEHEAD19 field sites to scale the LAD profiles. A sigmoid curve was





**Figure 3.** Scaled and re-sampled profiles used in the Child02 domain. Panels a and b show LAD profiles for the August IOP and panels c and d show profiles for the September IOP. Panels a and c show profiles for normal forests and panels b and d show panels for wetland forests. Note that there is no difference in the LAD profiles set up for normal and wetland forests during the August IOP simulations.

fitted to the leaf fall observations of each single tree. Afterwards, we grouped the plots into wetland and non-wetland plots and averaged the fitted sigmoid curves of all trees within each group as shown in Fig. 2 d, e. The resulting curves provide the percentage of fallen leaves. We then multiplied the LAD profiles in deciduous forest areas with the percentage of remaining leaves. For the mixed forest areas, we assumed that the reduction in LAI would be half that of the deciduous forest. This approach resulted in three different LAD profiles for the August IOP simulations before the beginning of leaf fall (broadleaf,





needle leaf, mixed) as shown in Fig. 3 a, b, and five LAD profiles for the September IOP simulations after the beginning of leaf fall (broadleaf, wetland broadleaf, mixed, wetland mixed, needle leaf) that are shown in Fig. 3 c,d. The LAD profiles were then assigned to different WISCLAND 2.0 land cover types (Table 2).

    Virtual tower measurements are conducted in the Child02 domain to be compared to actual field measurements at the EC stations. To create model conditions that are as similar as possible, the high-resolution profiles at these locations were not

adjusted to the mean value for the forest type, but were adjusted to the actual height of the vegetation at the respective station on a $100 \times 100$ m$^2$ area surrounding the virtual tower location.

### 3.3.3   Radiative Transfer Model

To simulate radiative transfer processes and their interactions with the plant canopy, the three dimensional and multi-reflective Radiative Transfer Model (RTM) offered within the PALM model system. The RTM calculates radiative fluxes and surface

net radiation including its components on the model geometry, which are then used to model the surface energy balance and evapotranspiration in the plant canopy. A detailed description and evaluation of the radiative transfer model is presented in Krč et al. (2021). In our model setup, the mean incoming surface short wave and long wave radiation values from the HRRR data over the Parent model domain was used as input to the RTM. This helps us to include effects of the spatially heterogeneous plant canopy and high clouds on the simulated surface radiation and flux budgets. The view factors and solar visibility for

surfaces were calculated using 10 horizontal discrete directions (azimuths) and 10 vertical discrete elevations. Plant canopy transmittance for each grid cell is calculated as a function of the prescribed LAD profiles as per Equation 90 in Maronga et al. (2020a) and stored as plant canopy sink factors for each grid cell. The number of reflection steps to be performed inside the RTM for the reflected short- and long-wave radiation between mutually visible surfaces were set at 2. Since the plant canopy is static, the view factors and canopy sink factors are precalculated before the atmospheric model time-stepping

starts. The precomputed view factors and canopy sink factors are applied to calculate irradiances and radiative heat fluxes at each radiation time step. After the configured number of reflections all the remaining radiative flux is considered as absorbed (Maronga et al., 2020a).

### 3.4   Virtual observational Infrastructure

Virtual tower sites were set up in the Child02 model at the same location as the 17 CHEESEHEAD19 tower sites. The volume

averaged turbulent time series data of $u,v,w,\theta,q$ were extracted at the model time step resolution at these locations. The data were extracted at vertical grid points corresponding to the field experiment tower measurement heights and also offset by one grid point offset to north-south and east-west. The virtual tower data were switched on after 05:00 CDT of the first simulation day to account for simulation spin up.

    Virtual flight tracks emulating the IOP airborne campaign flight tracks were simulated in the Child01 model for both August

and September test cases. Virtual flights are performed according to the methods described in Schröter et al. (2000) and Sühring and Raasch (2013). Virtual flights are effectively passive sensors that move along prescribed tracks, at prescribed heights with specific velocities. Flight heights were prescribed at 100 and 400 m above ground level, and speed was kept fixed at 82.31 m



$s^{-1}$ reflecting the field experiment transects. Virtual flights start at a waypoint, matching the field experiment location in space, move to their ending waypoint and then start back once they reach the end point. All flight transects start simultaneously, at
09:00 CDT of the first simulation day.

### 3.5 Data Analysis

In this manuscript, we focused on the simulated data from the Child01 model domain (3.2). The daytime data during 23 August and 24 September simulations were analysed, when all of the ABL was fully resolved within the Child01 model.

The ABL height field was calculated as the local maximum of the $d\theta(x,y)/dz$ following Sullivan et al. (1998). The domain
mean boundary layer height, $\langle z_i \rangle$ was then calculated from these 2D fields. We use the $z_i$ computed by this local-gradient method over the heterogeneous domain for scaling purposes, as suggested by MR13 Maronga and Raasch (2013).

## 4 Comparisons of Simulations With Field Measurements

As a first order fidelity check of our simulations of the field experiment days, we compare LES profiles and near-surface fixed-point time series with the field measurements. For profile comparisons, we present results for 23 August 2019 and 24
September 2019, when all of the CBL was fully resolved in the Child01 model in Sect. 4.1. For the August IOP simulation, instantaneous data were output at every 30 minutes and for the September IOP they were written out at every 15 minutes. For the profile comparisons, the LES data output closest in time with the radiosonde measurements were used. Some of the differences between the profiles can be attributed to this offset in time ($\sim$ couple of minutes). We compare the time series of $theta$, $q$, H and LE between the Child02 model and the CHEESEHEAD19 tower data. Table 3 presents summary statistics
for the comparisons between simulated and measured profiles and time series data.. For the profile data mean, median and Root Mean Squared Error (RMSE) of the deviations are reported. For the time series comparisons, their Pearson's correlation coefficient ($r$) and RMSE are reported. Only the daytime values of the heat fluxes were used for the calculation of summary statistics. All time mentions that follow refer to the corresponding simulation time.





**Table 3.** Comparisons between simulated and measured profiles of $\theta$, $q$ and time series of $\theta$,$q$, $H$ and $LE$. Intensity of the red highlight denotes higher differences.

| | | | $\theta$ (K) | | | | $q$ (g kg$^{-1}$) | | | | Daytime $H$ (W m$^{-2}$) | Daytime $LE$ (W m$^{-2}$) |
|---|---|---|---|---|---|---|---|---|---|---|---|---|
| | | Time (CDT) | 600 | 900 | 1300 | 1700 | 600 | 900 | 1300 | 1700 | | |
| Aug IOP | Profiles | Median of Absolute Difference | 1.66 | 1.21 | 0.61 | 0.46 | 0.62 | 0.48 | 0.43 | 1.23 | | |
| | | Mean Absolute Difference | 1.47 | 1.3 | 1.04 | 1.27 | 0.96 | 0.87 | 0.76 | 1.05 | | |
| | | RMSE | 2.71 | 2.06 | 1.4 | 2.62 | 1.16 | 0.92 | 0.72 | 1.23 | | |
| | Time series | Correlation Coefficient | 0.94 | | | | 0.49 | | | | 0.87 | 0.91 |
| | | RMSE | 2.74 K | | | | 5.39 g kg$^{-1}$ | | | | 48 | 60 |
| Sep. IOP | Profiles | Time (CDT) | 0645 | 0915 | 1300 | 1645 | 0645 | 0915 | 1300 | 1645 | | |
| | | Median of Absolute Difference | 1.45 | 1.66 | 1.58 | 0.38 | 1.29 | 1.19 | 0.55 | 1.52 | | |
| | | Mean Absolute Difference | 1.74 | 1.8 | 1.31 | 0.51 | 1.27 | 1.51 | 1.02 | 1.55 | | |
| | | RMSE | 2.08 | 2.13 | 1.45 | 0.63 | 1.48 | 1.82 | 1.59 | 1.64 | | |
| | Time series | Correlation Coefficient | 0.88 | | | | 0.96 | | | | 0.87 | 0.86 |
| | | RMSE | 2.8 K | | | | 7.19 g kg$^{-1}$ | | | | 49 | 59 |





## 4.1 Temperature and Moisture Profiles

**Figure 4.** Comparisons between PALM simulated (solid lines) radiosonde measured (dashed lines) and HRRR (dotted lines) profiles of theta (red) and water vapour mixing ratio (blue) for August and September IOPs. The shading around each simulated profile indicates the minimum-maximum range at each vertical level in the LES.





Radiosondes were released during the field experiment close to the US-PFa tall tower. Instantaneous LES profiles were extracted from the Child01 domain, at the location of the US-PFa tall tower. For both the August and September IOP simulations, PALM simulated early morning $\theta$ profiles showed lower near surface temperatures than the radiosonde profiles (Fig. 4). In their diurnal simulations using PALM Resler et al. (2021) and Gehrke et al. (2021) reported lower than observed near surface night-time temperatures and a more stable ABL than observations. Gehrke et al. (2021) discusses this issue, suggesting the role

of the SGS model and radiation scheme in combination with the grid resolution as well as the role of the LSM's surface energy balance parameterisation in combination with Monin-Obukhov Similarity Theory based computation of atmospheric fluxes at the first model grid point.

For the August IOP, the mean difference for the 1st 200 m between simulated and observed $\theta$ profiles at 06:00 CDT simulation time was 2 K, with a peak of $\approx 6$ K near the surface. At 09:00 CDT, the mean difference for the 1st 200 m became

3 K and near the surface $\approx 5$ K. Later during the day, the mean differences reduced substantially to 0.7 K at 13:00 CDT and 0.6 K at 17:00 CDT. Once the CBL was well mixed and fully developed in the afternoon, the simulations were able to capture the CBL height well. At 13:00 CDT, $z_{i\,simulated} \approx 1300$ m, while $z_{i\,observed} \approx 1500$ m. The differences between $\theta_{simulated}$ and $\theta_{observed}$ profiles at 13:00 and 17:00 CDT of the simulations arise mainly from their differences above the CBL, where the LES profiles were closer to the HRRR forcing data (Fig. 4). At 13:00, the mean absolute difference between the two below

1500 m was 0.5 K and at 17:00 CDT the mean absolute difference between the two below 1500 m was 0.3 K. Amongst all the four times comparisons were made, the mean absolute difference and the RMSE were the lowest at 1300 CDT at 1.04 K and 1.4 K respectively (Table 3).

The $q_{simulated}$ profiles for the fully developed CBL at 13:00 CDT on 23 August and for the evening ABL at 17:00 CDT followed the radiosonde measured profiles of $q_{observed}$. For the 13:00 CDT profiles, the RMSE between the two is 0.72 g kg$^{-1}$.

The differences largely occurred above the simulated CBL between 1250 and 1500 m where the simulated values are lower. At these heights, the radiosonde values were within the range of horizontal variability in $q_{simulated}$. The maximum horizontal variability for $q_{simulated}$ within the simulated CBL was at 13:00 CDT, with a mean range of 4 g kg$^{-1}$. Later during the day, at 17:00 CDT the simulations were drier, with a RMSE of 1.23 g kg$^{-1}$.

The September IOP simulations showed the same patterns as the August IOP simulations for $\theta_{simulated}$ and $q_{simulated}$ pro-

files when compared with the radiosonde measured profiles. The early morning near surface $\theta_{simulated}$ values were smaller, while the fully developed mixed layer values later agreed well. The mean absolute difference for the 1st 200m between $\theta_{simulated}$ and $\theta_{observed}$ profiles at 06:45 CDT was 4.2 K with a maximum of 7.6 K at 30 m. At 09:15 CDT the mean absolute difference for the 1st 200m between $\theta_{simulated}$ and $\theta_{observed}$ profiles was 4.25 K. Later during the day, at 13:00 CDT $\theta_{simulated}$ and $\theta_{observed}$ had mean absolute difference and RMSE of 1.31 K and 1.45 K and at 16:45 CDT they reduce to their

lowest at 0.51 K and 0.63 K respectively and remain nearly constant throughout the model vertical extent. The spatial variation of $\theta$ profiles inside Child01 model was lower for the September IOP simulations than the August IOP simulations, as seen in the smaller extent of the shaded minimum-maximum ranges. Most of this variation occurred in the 1st 100 m at 06:45 CDT and 16:45 CDT and within the 1st 200 m at 09:15 CDT. The maximum spread was seen at 1300 CDT which $\approx 4$ K.



The $q_{simulated}$ values for September IOP, corresponded well with $q_{observed}$ during the morning hours at 06:45 CDT and
09:15 CDT. They diverged higher up from the surface, above around 600 m following the shape of the input HRRR forcing
$q$ profiles. $q_{simulated}$ and $q_{observed}$ profiles at 13:00 were very close to each other within the simulated CBL ($\approx 800m$) with
a mean absolute difference of 0.39 g kg$^{-1}$. The radiosonde data showed an advection event centred around 1180 m which
was not seen in the simulated profiles. However this spike fell within the spatial variability captured at the same height in the
$q_{simulated}$ profiles. Similar to the August IOP, the 16:45 CDT values were very similar in shape but $q_{simulated}$ values were
lower with a RMSE of 1.64 g kg$^{-1}$. At 13:00 CDT and 16:45 CDT the simulated spatially local values were higher than the
HRRR forcing profiles (Fig. 4) but the domain averaged values are closer to each other (Fig. 5).



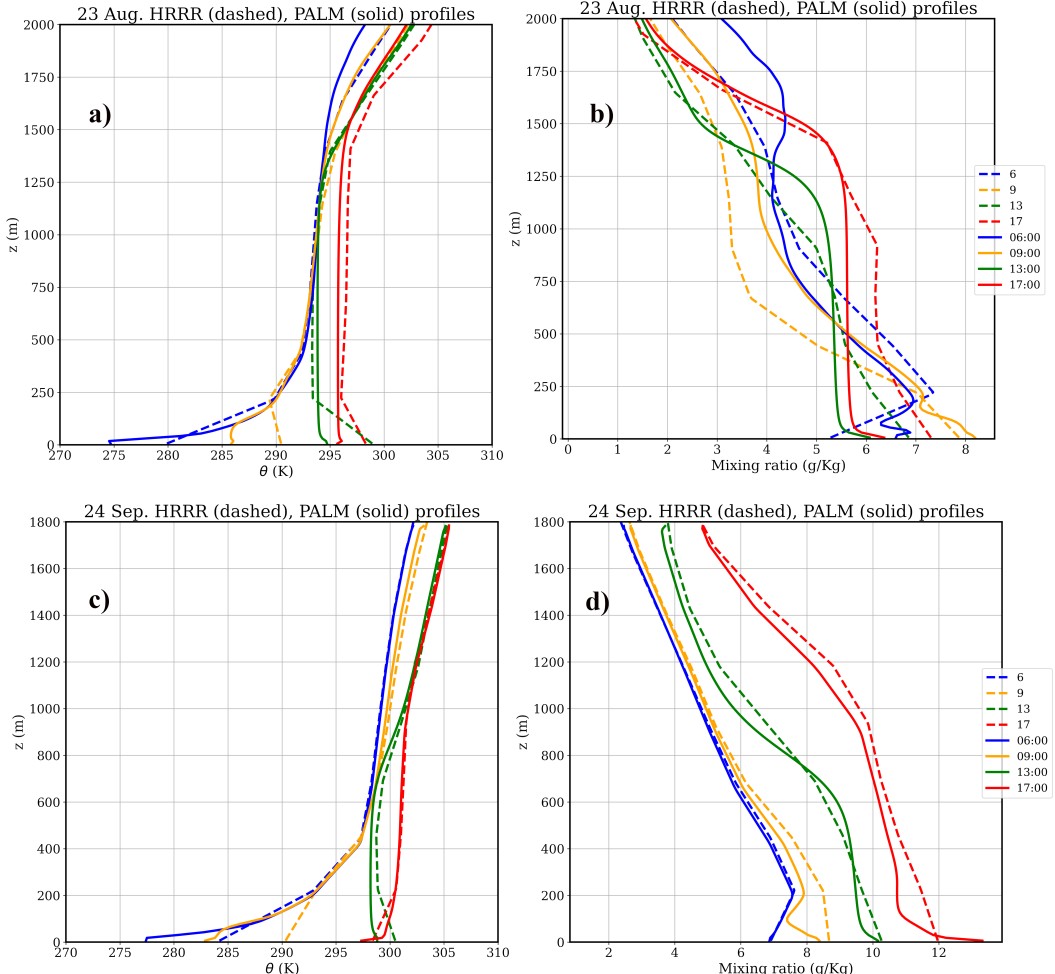

**Figure 5.** Comparisons between PALM simulated and input HRRR profiles of $\theta$ (a,c) and water vapour mixing ratio (b,d) for August (a,b) and September (c,d) IOPs. Different colours are used to distinguish between differing times as shown in the legend for each IOP test case. Dashed lines indicate HRRR profiles and solid lines indicate PALM simulated domain-averaged, instantaneous profiles.

## 4.2 Horizontal Wind Profiles

Figures 6 and 7 present the simulated, radiosonde measured and input HRRR profiles of horizontal winds, $u$ and $v$ for the 23 August and 24 September simulations respectively. The simulated profiles presented alongside radiosonde profiles are ensemble averages of fixed-point, instantaneous LES data, similar to the simulated $\theta$ and $q$ profiles presented earlier. The simulated profiles presented alongside the HRRR data are domain averaged instantaneous data. For these wind profiles, the shading indicates the standard deviations of velocity magnitudes at each heights across ensembles as indicators for the spread





of simulated values. The ensemble standard deviations were chosen instead of the spatial range of velocities, since the spatial range of wind velocities would be a less physically meaningful metric for model data comparisons.

In the 23 August simulations, the simulated winds were of the same order of magnitude as the measured wind profiles. The exact values of simulated and measured afternoon wind profiles at 13:00 and 17:00 CDT also oscillated around each other. The simulated domain mean wind profiles were closer to the input HRRR forcing profiles (Fig. 6 c, d). More so, for the afternoon profiles at 13:00 and 17:00 CDT and for heights inside the ABL, at z ≤ 1500 m.

       Simulated and measured wind profiles for the 24 September test case also showed the same patterns as the summertime
simulations. The boundary layer winds were also higher in the wind shear driven early autumn September boundary layer. The simulated and radiosonde measured profiles were close to each other and vary around each other (Fig. 7 a, b). The domain mean simulated wind profiles follow the forcing HRRR profiles very closely (Fig. 7 c, d) for all the time steps compared.

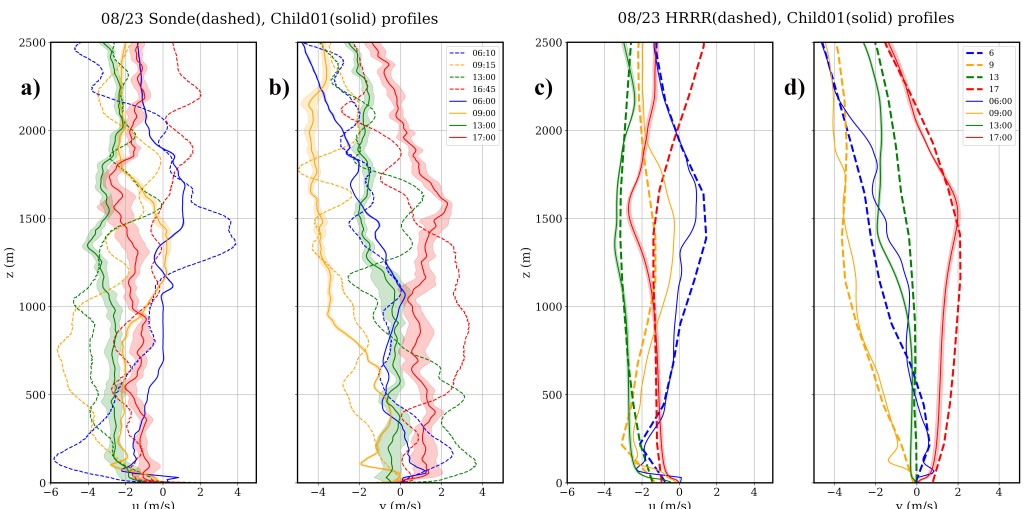

**Figure 6.** Panels a) and b) present comparisons between simulated (solid line) and radiosonde measured (dashed line) profiles of u and v wind respectively. Ensemble mean values of simulated profiles are presented. Shading represents the standard deviation across ensemble members. Panels c) and d) present comparisons between domain mean simulated (solid line) and the input HRRR (dashed line) profiles of u and v wind respectively. Different colours are used to distinguish between differing times as shown in the legend. Shading represents the standard deviation across ensemble members.





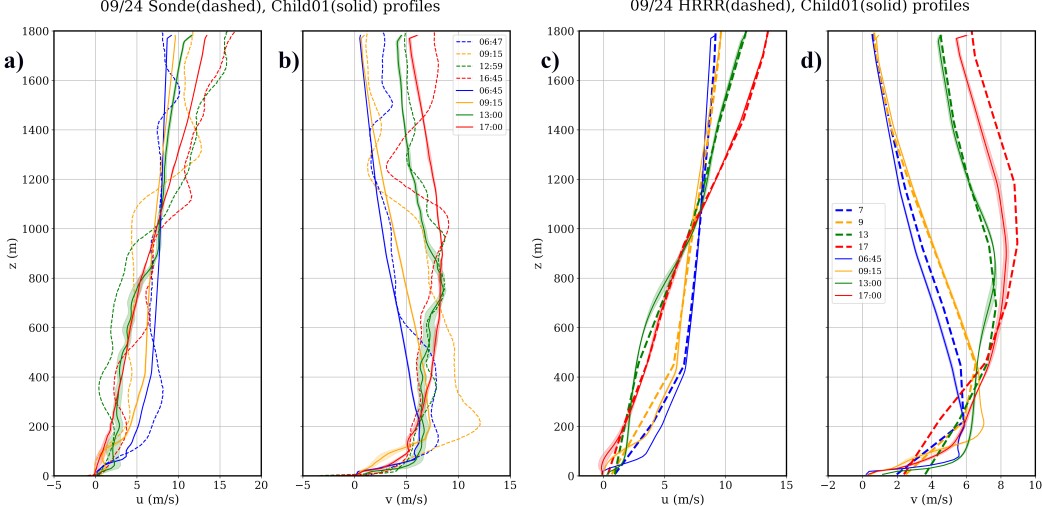

**Figure 7.** Same as Fig. 6 but for 24 September simulations.

## 4.3 Near Surface Temperature and Moisture Time Series

Above-canopy time series of simulated $\theta$ and $q$ values were extracted at $z = 32$ m from Child02 model for 12 CHEESE-HEAD19 tower locations, where the field experiment measurement heights were also 32 m (Butterworth et al. (2021) supplement). The ensemble mean time series values across locations were compared between the simulations and tower data, along with the time series values from the input HRRR forcings data at the same grid level (Fig. 8). 30-minute averaged values for August and September IOPs are presented. Shadings denote one standard deviation



**Figure 8.** PALM simulated, CHEESEHEAD19 tower measured and input HRRR forcings data time series for temperature (a,b) and moisture (c,d) for the August (a,c) and September (b,d) IOP simulations.

The $\theta_{simulated}$ time series showed good agreement against the tower measured values for both days of the August and September IOP test cases, with a correlation coefficient of 0.94 for August and 0.88 for September days (Fig. 8, Table 3). The $\theta_{simulated}$ values in August have higher peaks and lower lows, resulting in a RMSE of 2.74 K with respect to mean $\theta_{observed}$ at the towers. For 22 August the maximum $\theta_{simulated}$ was 295 K at 15:30 CDT while the maximum tower measured $\theta$ value for 22 August was 291 K at 14:30 CDT. Likewise, for 23 August the maximum value was 296 K at 18:00 CDT and the maximum $\theta_{observed}$ was 293.5 K at 17:00 CDT. The $\theta_{simulated}$ values were closer to the forcing data.





The September IOP simulation is able to capture the morning warm up related to CBL initiation for the first day (Fig. 8 b). The simulations miss the rainfall induced dip during 24 September night at 21:30 as seen in the tower data although the cooling of the stable boundary layer is captured well, as it was reflected in the HRRR data as well. However, the $\theta_{simulated}$ peak for 25 September is higher, at 294.7 K, than the tower measured value of 290 K.

        The simulations were dry-biased for both the August and September IOP days, with a RMSE of 5.39 g kg$^{-1}$ in August and
7.19 g kg$^{-1}$ in September with respect to tower measured values. Even though RMSE does not provide any directionality for the error, we use it here for the sake of consistency with the rest of the analysis. The simulated $q$ values were of the same range of magnitudes as the forcing HRRR data, which had lower magnitudes than tower measured values.

        For the August IOP, the intra-day variability in $q_{simulated}$ resembled the tower measured data, where all three peaks as seen in the tower data were simulated at nearly the same times (Fig. 8 c). However, there was a time lag between the simulated and
tower measured values, of the order of a couple of hours, because of which their correlation coefficient was only 0.49.

        Likewise, for the September IOP the low frequency variations of the $q_{simulated}$ values and the tower measured values were similar, approximating a Gaussian, with the $q_{simulated}$ values modulated by the HRRR values in amplitude and spread (Fig. 8 d). The $q_{simulated}$ peak was around 12.5 g kg$^{-1}$ at 24 September 18:30, while the tower measured values showed almost double the same value at 22.5 g kg$^{-1}$. The simulated values were also smoother than the tower measured values for the September
simulations, especially for 25 September, indicating lower spatial variance in the simulation.

### 4.4    Near Surface Wind Time Series

We present a comparison of the simulated near-surface wind time series from the Child02 model domain and the CHEESE-HEAD19 tower measured wind time series along with the input HRRR forcing data at the same model vertical grid level for the two IOP test cases in Fig. 9.



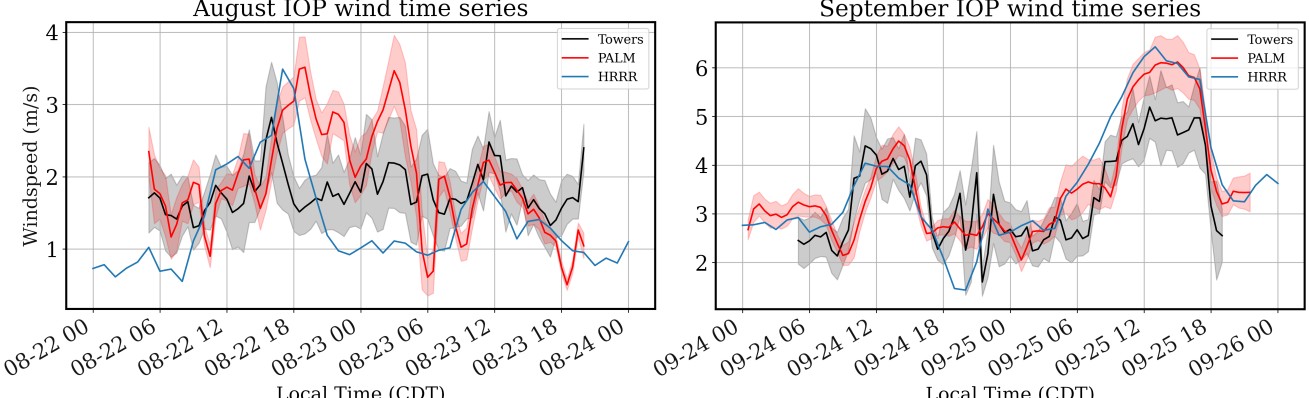

**Figure 9.** PALM simulated, CHEESEHEAD19 tower measured and input HRRR forcings data wind time series. Above-canopy time series of simulated wind were extracted at $z = 32$ m from Child02 model for 12 CHEESEHEAD19 tower locations, where the field experiment measurement heights were also 32 m. The simulated ensemble mean time series values across locations are compared against the mean tower measured data, along with the time series values from the input HRRR forcings data at the same grid level. Half-hourly averaged values for August and September IOPs are presented. Shadings denote one standard deviation.

For the August simulations, the simulated and tower measured value magnitudes showed good agreement. The mean simulated and tower measured daytime (06:00 - 20:00 CDT) winds during 22 August were 2.3 $\mathrm{ms^{-1}}$. For 23 August, the mean tower measured winds were $1.8\mathrm{ms^{-1}}$ from 06:00 - 18:00 CDT, while the mean simulated values were lower at 1.5 $\mathrm{ms^{-1}}$. The simulations picked up the spike in the wind speed seen in tower measurements around 22 August 16:00, but the modelled spike occurred later around 18:00 CDT. Simulations for 23 August were able to pick up the afternoon decrease of near surface winds

from 12:00 - 18:00, with the same order of magnitude as the boundary layer becomes more convective. The time series comparisons indicate that the PALM simulated mean winds for the August IOP are representative of the free convective conditions observed during the field experiment.

     In the September IOP test case, when surface winds drive boundary layer turbulence, the simulations picked up the diurnal cycle and magnitudes of the tower measured near surface winds well. Simulated winds were higher on the second day of

25th September at 6 $\mathrm{ms^{-1}}$ and closer in magnitude to the HRRR forcing data while tower measured mean wind speed was at 5 $\mathrm{ms^{-1}}$. The measured winds during 24 September night, from 1800 till midnight showed much more variability than the simulations.

### 4.5   Near Surface Heat Flux Time Series

The near surface simulated $H$ and $LE$ time series were compared with the field experiment measurements similar to the wind

and scalar time series (Fig. 10). Since the maximum canopy height in the domain was around 32 m, only those tower sites with measurement heights $\geq 32$ m were considered for comparisons. The simulated mean daytime $H$ time series values across CHEESEHEAD19 tower locations were greater than the tower measured values for both the August days, with a RMSE of



48 Wm$^{-2}$. The differences were the greatest on 22 August, when the RMSE of simulated and tower measured values was 64 Wm$^{-2}$. The tower measurements of incoming short wave radiation during 22 August showed an appreciable influence of

cloud cover compared to those measured on 23 August. The tower mean measurements on August 22nd showed a maximum value of 577 Wm$^{-2}$, while the input HRRR surface forcing values had a maximum of 705 Wm$^{-2}$. While for 23rd August the mean tower measured short wave incoming radiation was 888 Wm$^{-2}$ and the domain mean HRRR input values at the surface were close at 855 Wm$^{-2}$. For the 22nd August simulations the input surface values for the Parent domain from the HRRR data did not reflect the decreased available incoming short wave radiation and this could have contributed towards the

simulations having higher than observed $H$ partitioning. Despite the offset in the first day of simulations, the simulated and measured daytime $H$ time series for both the simulation days showed a linear relationship with a correlation coefficient of 0.87 (Fig. 10b, Table 3). For the 23 August simulations the simulated daytime $H$ magnitudes were closer to the tower measured values with a RMSE of 23 Wm$^{-2}$.

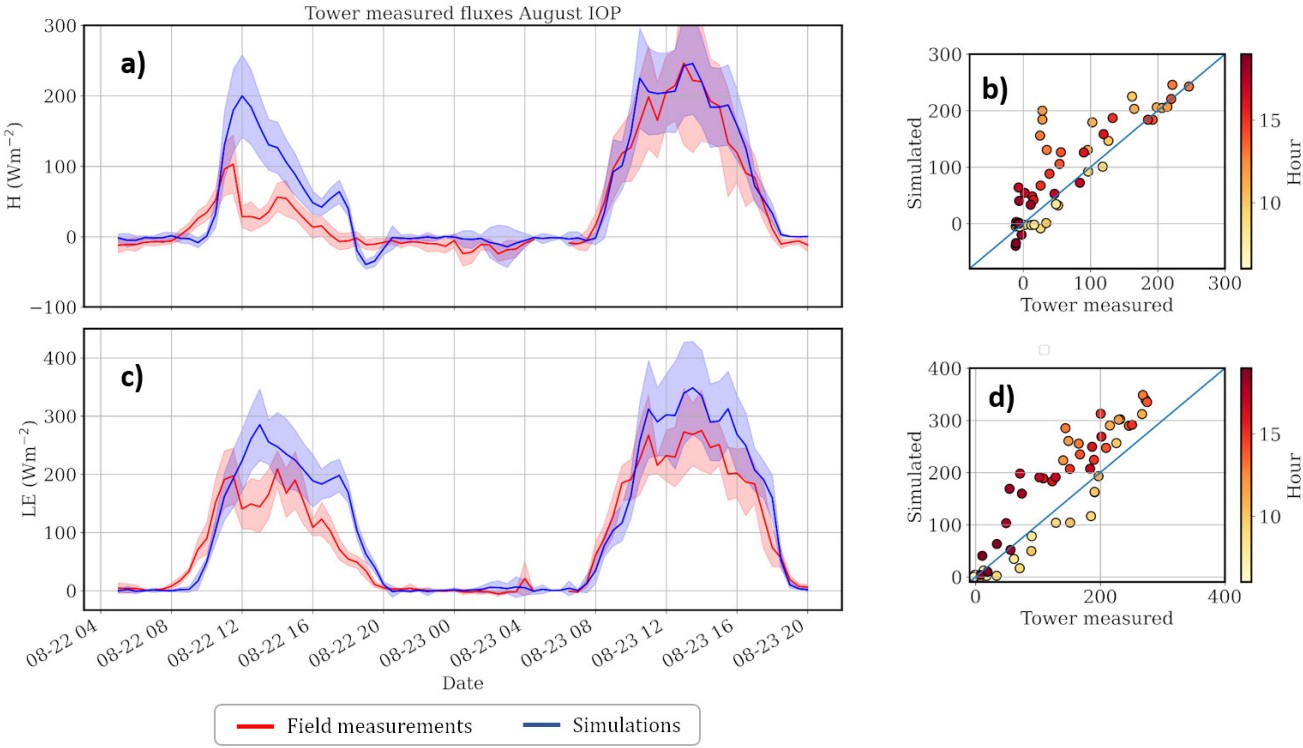

**Figure 10.** Simulated vs observed near surface turbulent fluxes at the CHEESEHEAD19 tower locations for the August IOP from Child02 model. Mean flux time series values across 12 tower sites with measurement heights $\geq 32$ are presented. CHEESEHEAD19 flux towers in red, simulated values in blue. For LES, the mean across 8 ensembles, and all 12 towers is presented. Shading shows the standard deviation across the hole data; b, d : Scatter plots between the simulated and tower measured daytime (0600 - 2000 CDT) mean $H$ and $LE$ for 08/22 and 08/23.





The 22 August simulations also showed a delay in the CBL initiation of the order of an hour, seen in both the $H$ and $LE$ time

series. This could be a result of the idealised soil model set up, with uniform soil properties throughout the domain. This delay is not seen for the second day, when both simulations and observations showed the initiation of surface driven CBL growth at 08:00 CDT.

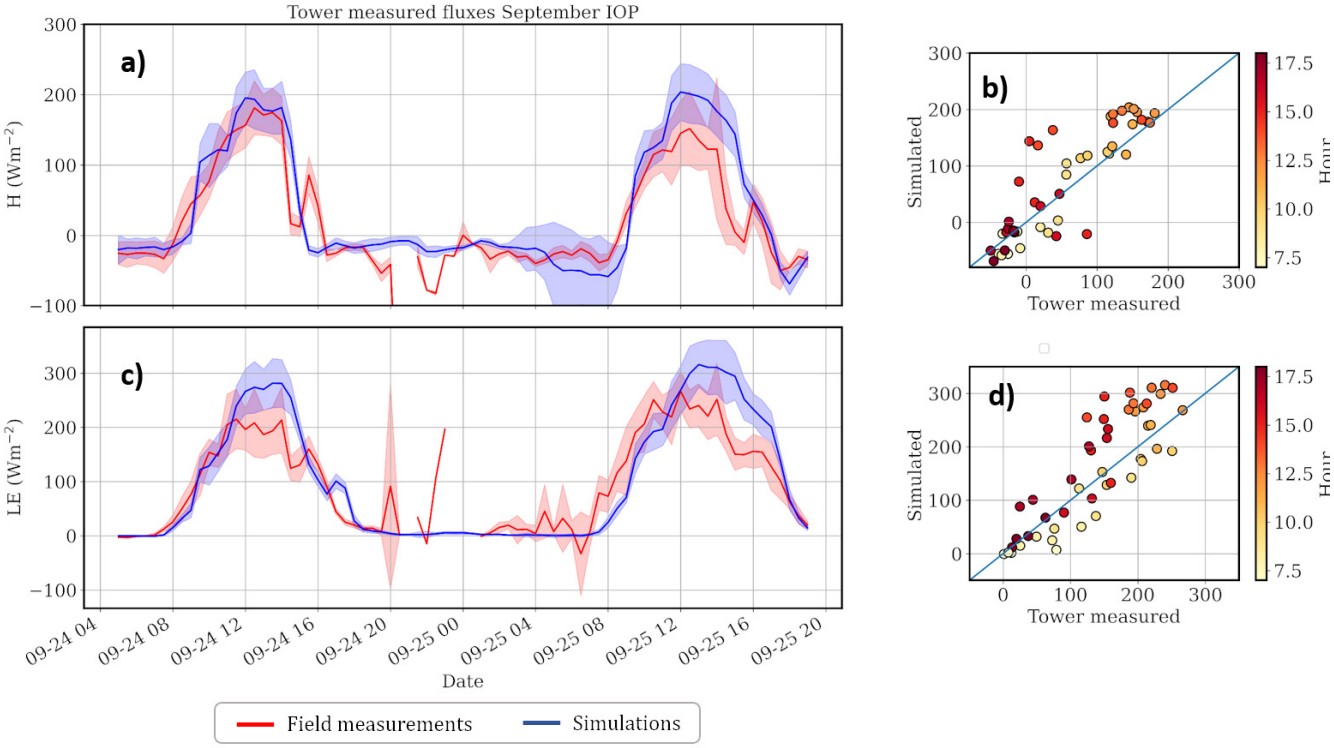

**Figure 11.** Same as Fig. 10, but for September IOP. Daytime hours 07:00-18:00 CDT.

The simulated $H$ time series on 22 August had a steep increase in the morning that peaked at 200 $\mathrm{Wm}^{-2}$ at noon while the tower measured $H$ peaked at 100 $\mathrm{Wm}^{-2}$ at 11:30 CDT. Similarly, the simulated $LE$ had a higher peak of 285 $\mathrm{Wm}^{-2}$ than the

tower measured value of 200 $\mathrm{Wm}^{-2}$ and remained consistently higher through afternoon till the CBL collapsed at 20:00 CDT. For 23 August (the second day) of simulations, the simulated $LE$ magnitudes remained higher than tower measured values for most of the day. The simulated and tower measured daytime $LE$ time series are linearly well correlated with a correlation coefficient of 0.91.

During the stable boundary layer of 22 August evening till 23 August morning, the mean $H$ measurements showed small

negative heat fluxes from 17:00 CDT 22 August till 04:30 CDT 23 August. The simulated $H$ values at night-time showed only one below 0 excursion lasting about 2 hours, with a minimum of -39 $\mathrm{Wm}^{-2}$ and then neared zero values for the rest of the night. The mean tower measured values while they were never as low at night in the stable boundary layer were consistently negative, with a mean value of -12 $\mathrm{Wm}^{-2}$ and a minimum of $-25\mathrm{Wm}^{-2}$. This difference could also have contributed towards near



surface $\theta_{simulated}$ being lower than radisonde measured $\theta_o bserved$, since a lot of heat was not removed from the atmosphere
to the land surface in the simulations.

# 5    Atmospheric Boundary Layer Evolution and Organisation

## 5.1    Temporal evolution

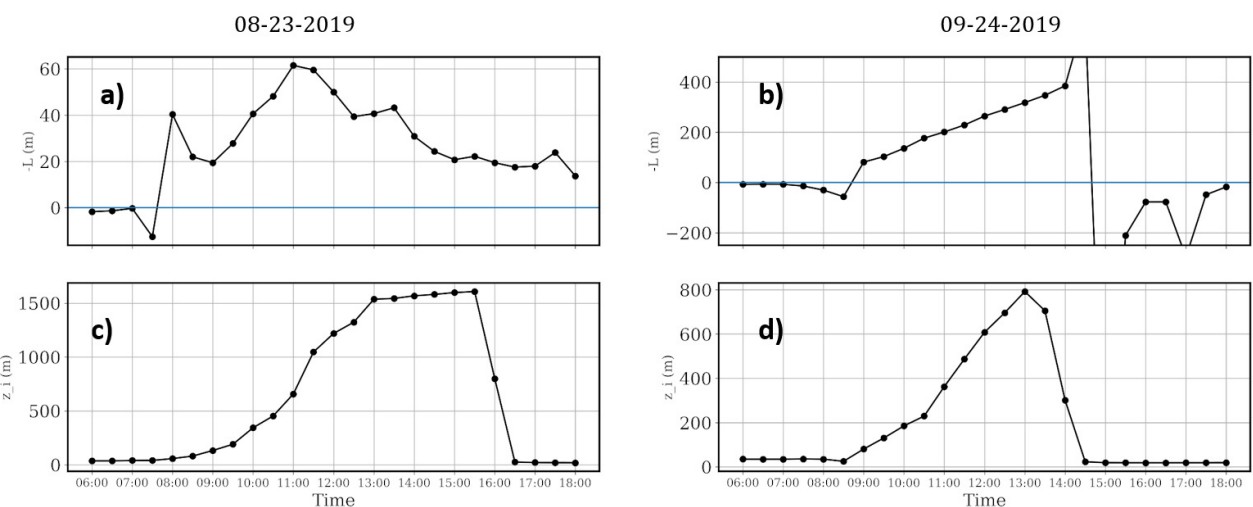

**Figure 12.** 30-minute, horizontal domain-mean Obukhov length (a,b, $-L$ presented) and $z_i$ (c,d) for 08/23 (a,c) and 09/24 (b,d) runs from Child01 model.

For the 23 August and 24 September day-time simulations we looked at the ABL evolution and compared their wind shear vs buoyancy driven nature, using data from the Child01 model. The domain-mean Obukhov length (Obukhov, 1946; Monin
and Obukhov, 1954) was calculated for the Child01 model for 30-minute intervals as $L = -u_*^3\overline{\theta_0}/kgQ_0$. Here, $u_*$ is the above-canopy friction velocity, calculated at 36 m as the square root of resolved scale shear stress $[(\langle\overline{u'w'}\rangle)^2 + (\langle\overline{v'w'}\rangle)^2]^{1/4}$; $g/\overline{\theta_0}$, the buoyancy parameter where $g$ is the gravitational acceleration and $\overline{\theta_0}$ the horizontal domain mean above-canopy potential temperature at 36 m; $k$ the von Kármán constant = 0.4 and $Q_0$ the calculated above-canopy kinematic vertical heat flux $\langle\overline{w'\theta'}\rangle$ at 36 m. Since $L$ has units of length, a non-dimensional stability parameter $-\frac{z_i}{L}$, can be defined with values close to 0 indicating
a statically neutral, wind shear-driven, forced convective ABL and as the value increases, the ABL becomes more buoyancy-driven, free convective and statically unstable (Stull, 1988; Moeng and Sullivan, 1994).

On 23 August from 08:00 - 16:00 CDT as the simulated day-time ABL grew and decayed, 30-minute $-L \in [20, 60]$ m (Fig. 12 a). The fully developed CBL had domain-mean, 30-minute $z_i = 1500$ m, which was also the ABL height seen in the radiosonde measured $\theta$ profile at 13:00 CDT on 23 August (Fig. 4). For the 24 September simulations, from 09:00 - 14:00 CDT



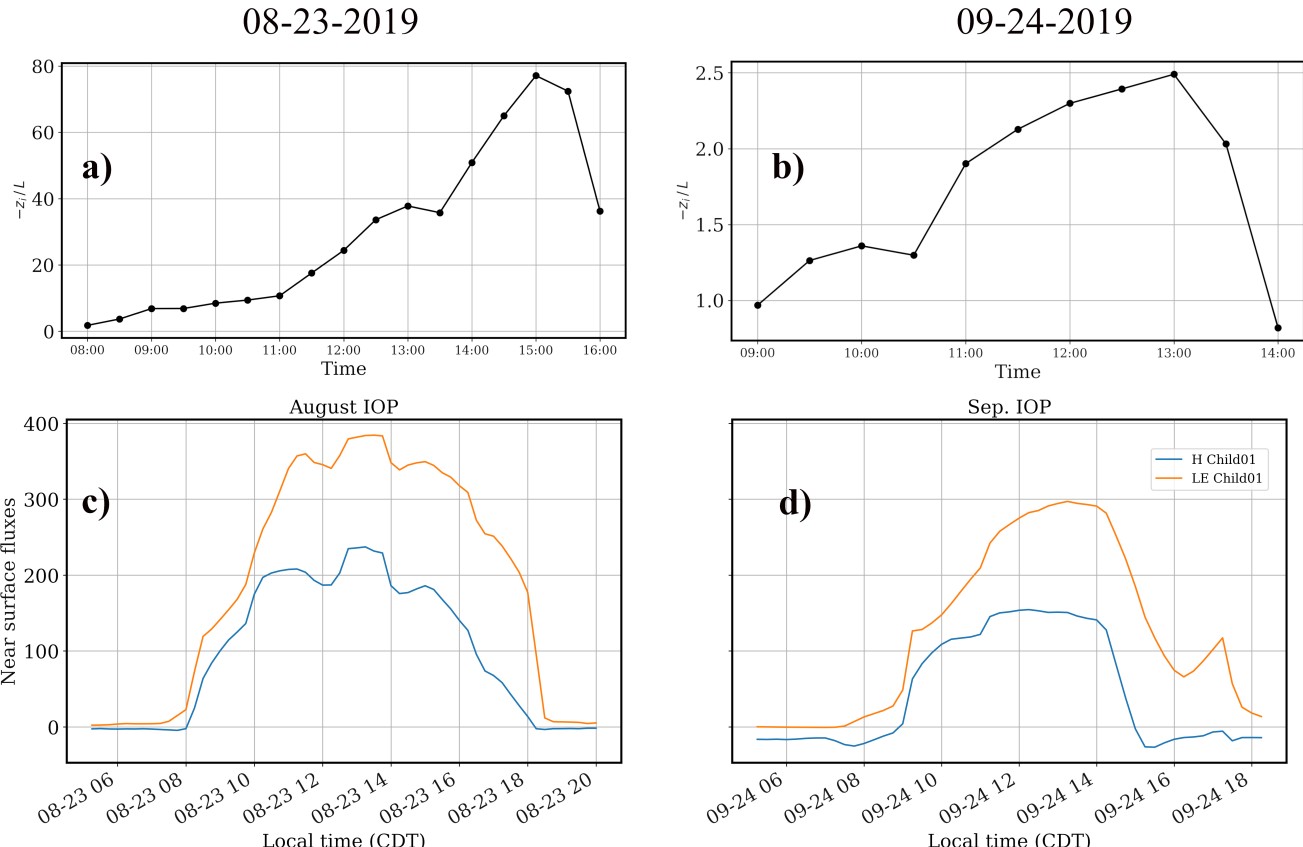

**Figure 13.** a,b : daytime 30-minute $-\frac{z_i}{L}$ plots for 08/23 and 09/24 Child01 simulations. Simulated, domain averaged above-canopy heat flux time series for 08/23 (c) and 09/24 (d) at 36 m above surface from Child01 model.

as the ABL evolved, 30-minute $-L \in [81, 385]$ m, representative of the stronger mean winds. The simulated ABL was also shallower than the 23 August simulations, with maximum $z_i = 800$ m at 13:00 CDT (as was also seen in the profile comparisons at 13:00 24 September in Fig. 4).

From 30-minute time series data for the 23 August simulations, $-z_i/L \in [1.8, 40]$ m for 08:00-13:00 CDT and later in the afternoon $-z_i/L \in [40, 80]$ from 13:30-16:00 CDT. The simulated domain mean total surface heat flux for 10:00-16:00 CDT

was 535 $\mathrm{Wm}^{-2}$ (Fig. 13 c) with a domain-mean above-canopy wind speed of 2.25 $\mathrm{ms}^{-1}$ (Fig. 14). The intra-day variability and range of values of $-z_i/L$ for the simulated ABL for 23 August indicate a more wind shear driven forced convective ABL in the early morning hours (08:00 - 11:00 CDT) as the mixing layer starts to grow and a free convective boundary layer later during the day that grows and fully develops to 1500 m by 13:00 CDT.

The range of values of $-z_i/L$ for the 24 September simulations were smaller than the 23 August simulations, with $-z_i/L \in$

$[0.4, 2.4]$. The domain and time mean daytime surface heat flux from 10:00 to 16:00 CDT at 36 m was 326 $\mathrm{Wm}^{-2}$ (Fig. 13 d)



which was 209 Wm$^{-2}$ lesser than the 23 August simulation. The domain mean boundary layer winds were also stronger, ranging from 4.5 ms$^{-1}$ at 11:00 CDT to 7 ms$^{-1}$ at 13:30 CDT (Fig. 14). These values indicate that the 24 September simulations have a wind-shear driven, forced convective ABL from 09:00 to 14:00 CDT, with a peak $z_i$ of 800 m around 13:00 CDT.

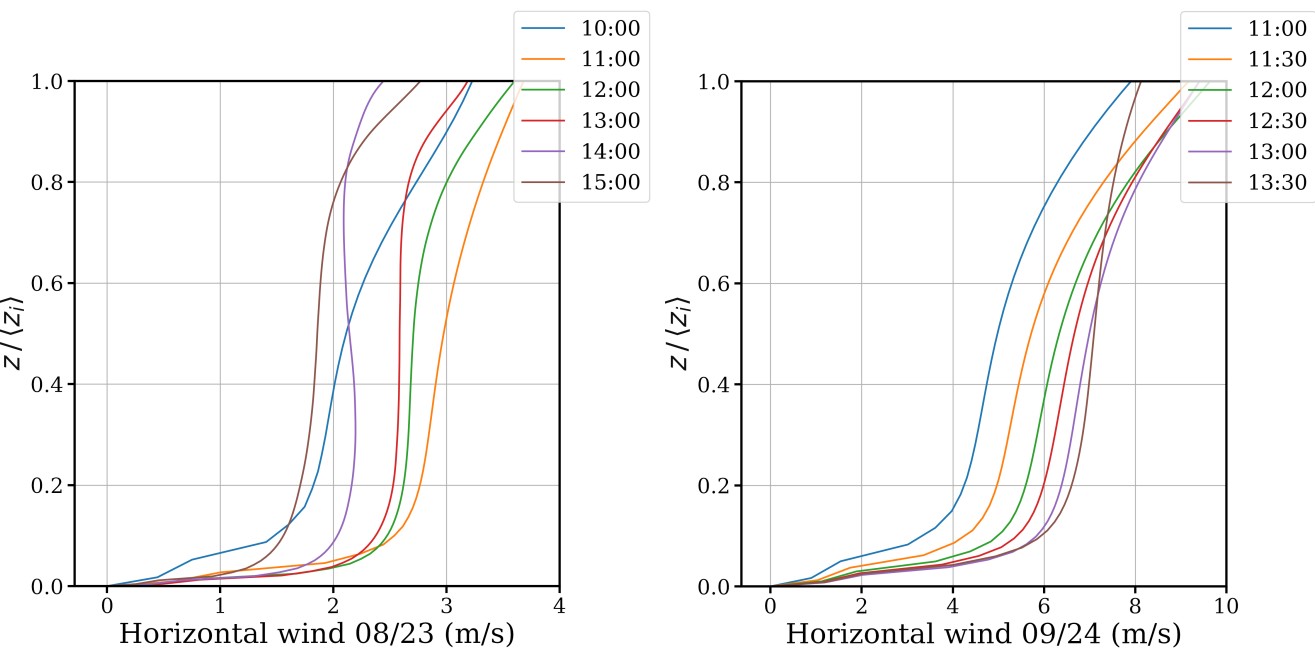

**Figure 14.** Simulated domain-mean horizontal-wind profiles for 23 August and 24 September. 30-minute averaged profiles for different times shown in different colours.

## 5.2 Horizontal evolution

Figure 15 shows the time-ensemble averaged $z_i$ horizontal cross-sections normalised by $\langle z_i \rangle$ for the Child01 model. The cross-sections were produced using 30-minute averages and 8 ensemble runs.

On 23 August morning 11:00 CDT, the boundary layer winds were easterly and the normalised $z_i$ cross-section (Fig. 15 a) showed signals of roll like convection that were oriented West to East of the model domain. The simulated domain mean $z_i = 657$ m with a spatial maximum of 985 m. Strong North-South spatial gradients were visible to the North-West of the

model domain. These were signals of the roll like convection with influence from the lake situated at the northern edge of the domain. Smaller than domain mean $z_i$ values were also seen near the lake, due to colder temperature of the lake surface. ABL heights were also shallower at the southern edge of the domain, (around $y \in [0, 5000]$ m & $x \in [5000, 20000]$ m).

In the afternoon, the ABL becomes free convective (Fig. 13 ) and fully developed at 14:00 CDT (Fig. 15 b). The simulated domain mean $z_i = 1568$ m with a spatial maximum of 1682 m. The CBL had lower spatial gradients, with most of the spatial

variations within $0.88\langle z_i \rangle$-$1.12\langle z_i \rangle$. Within this range, the CBL had a gentle east-west gradient, with higher values to the west



of the model domain. The lower than domain mean $z_i$ around the lake to the north can also be seen in the 14:00 CDT $z_i$ horizontal cross-section, however the simulated spatial gradients are smoother than the morning ABL at 11:00 CDT.

$z_i$ horizontal cross-sections of 24 September simulations for a forced convective ABL (Fig. 15 c) did not show signals of roll-like convection as distinctly as in the ensemble and time averaged $z_i$ plots of 23 August 11:00 CDT simulations. The

boundary layer heights were also shallower than the convective boundary layers during the August simulations. The simulated domain mean $z_i = 382$ m with a spatial maximum of 528 m. Mean boundary layer winds $\approx 6ms^{-1}$ were blowing in from the south and the shallower $z_i$ at the southern edge ($x \in [5000, 14000]$ m) were elongated along-wind.

In the afternoon, at 13:00 CDT, the $z_i$ spatial variations occurred at larger scales but around the same range of $0.8\langle z_i \rangle$-$1.2\langle z_i \rangle$. The simulated domain mean $z_i = 845$ m with a spatial maximum of 986 m.






**Figure 15.** x-y cross sections of time-ensemble averaged atmospheric boundary layer height, normalised by the domain mean, for 08/23 (a) 11:00 CDT and (b) 14:00 CDT; 09/24 (c) 11:00 CDT and (d) 13:00 CDT.

## 6  Discussion

Our null hypothesis was that LES, initialised with real-world surface heterogeneity, develop similar mesoscale structures and patterns as observed in reality. To include the changing synoptic scale forcings over the study domain, we used the hourly





NCEP-HRRRv4 reanalysis data product as the LES boundary conditions (Sect. 3.1). This inclusion had an influence on the simulated mean quantities (Fig. 5 and 8, Table 3) , especially the near surface moisture values, which were lower than the
measured mixing ratio values during the IOPs. However, the offset magnitudes were nearly constant throughout the simulations. For the above-canopy $q$ time series, simulations were lower than observations for the August IOP runs with RMSE = 5.59 $\mathrm{gkg}^{-1}$ and for September IOP runs the RMSE = 7.19 $\mathrm{gkg}^{-1}$. As simulations of field experiment days, the simulated mean atmospheric boundary layer state variables were well correlated with the radiosonde and tower measurements (Sect. 4 ). The simulated above-canopy daytime heat fluxes were also of the same order of magnitude as the tower measured values during
the field experiment, with a mean RMSE between observations and simulations of 54 $\mathrm{Wm}^{-2}$ for both the IOPs (Table 3). The above-canopy simulated daytime heat fluxes for 23 August and 24 September at the CHEESEHEAD19 tower locations had a Pearson correlation coefficient ($r$) of 0.9 with the concomitant tower measured values. They also showed similar diurnal variations with respect to ABL initiation and collapse (Fig. 10, Fig. 11). This gives us confidence that our model results simulate the CHEESEHEAD19 IOP daytime boundary layers and can be used to evaluate mechanisms that generate surface-
heterogeneity induced mesoscale circulations in the diurnal ABL.

## 6.1  On Large Eddy Simulations of Targeted Field Experiments

LES has been an effective tool to simulate the observed ABL mean properties and turbulence characteristics during targeted field campaigns and experiments. One of the first attempts was by Hechtel et al. (1990), who simulated an afternoon CBL of the Boundary Layer Experiment 1983 (BLX83), with synthetic surface sensible and latent heat flux fields generated to match
aircraft measured surface skin temperature power spectra. The heat flux amplitudes were set to vary along the diurnal cycle. The synthetic fields contained model grid size (150 m) variations as well as coherent areas with lengths of the order of 450 - 900 m. They performed a control run with homogeneous surface fluxes as well and compared both with field measurements. They reported no significant differences between the two runs in mixed layer development and area averaged statistics. For the heterogeneous case, their simulations showed no clear evidence that convective thermals were preferentially localised over
surface gradients. They proposed model resolution and inclusion of an imposed mean geostrophic wind $\approx 7\ \mathrm{ms}^{-1}$ as possible reasons for the lack of sensitivity to non-homogeneous surface boundary conditions. In our simulations, we find that the heterogeneity effects are stronger during the summer time runs, with weaker mean winds (not shown here) and much weaker for the September runs with mean winds $\approx 6\ \mathrm{ms}^{-1}$, in-line with their hypothesis.

LES of field campaign days coupled with a land surface model were later performed , to better represent the observed land-
atmosphere feedbacks. Given the multi-dimensional surface heterogeneity of our study domain, occurring due to the gradients of properties in plant canopy and land surface classes we could not performing a representative homogeneous control using a homogeneous land surface or plant canopy. However, investigators who have looked into the role of a LSM coupled to PCM, even as simple as two-source models for the land surface have reported better agreements between simulated and tower measured near surface heat fluxes.

Albertson et al. (2001) performed coupled LES of one day of the Monsoon '90 experiment, with periodic boundary condi- tions and time varying surface fluxes, computed from remotely sensed land surface properties and simulated atmospheric state





variables. They used a two-source model to separately account for the contributions from bare soil and vegetation within each model grid cell for their study area, with a wide range of fractional vegetation cover. Their setup, in theory, was similar to the coupled CHEESEHEAD19 LES framework, that used a LSM and PCM to resolve the surface atmospheric coupling over the
forested domain.

The spatial averages of sensible and latent heat fluxes from their simulations were within 10% of the tower measured values during the field experiment. However, the tower measured $H$ was computed using flux variance method (Tillman, 1972)) based on empirical formulations between air temperature, standard deviation and $H$ in the Monin-Obukhov Similarity Theory framework. Then $LE$ was computed as the residual of the surface energy balance. They compared their estimates of regional
averages with 1D EC technique and found their estimates to be within 20% of the measured values. The locally simulated surface flux values in our simulations have a comparable or better match to the tower measured fluxes.

Later investigators started exploring surface-atmospheric coupling using fully coupled land surface models and for the diurnal ABL. In their coupled LES study Huang and Margulis (2010) simulated one day (06:00 - 18:00 CST) of the SMACEX 2002 field experiment. Their simulated domain mean surface fluxes and profiles agreed well with tower and radiosonde measured
values but they reported no significant difference in domain mean surface fluxes, between fully coupled and un-coupled simulations. They proposed that this could be because the differences over the two dominant vegetation types cancel each other, and moreover the analysis was only performed for 30-minute averaged data at 13:00 CST. In line with the previous LES studies (Albertson et al., 2001; Bertoldi et al., 2007; Huang and Margulis, 2009), they suggested that surface-atmospheric coupling dampens the amplitude of the simulated heat fluxes, with simulated sensible heat fluxes reaching as low as $\approx 18\%$ lower in the
fully uncoupled case. However, the land surface class distribution in their simulations is essentially very similar to an idealised 2 dimensional heterogeneity, with only the short canopy of soy and corn crops dominating.

Nevertheless, the RMS Deviation of their flux time series is of the same order of magnitude as ours, with 30 W m$^{-2}$ for $H$ and around 41 W m$^{-2}$ for $LE$ at the footprint scale. Their simulated and observed air temperature values agree well, with $\rho = 0.97$. The simulated moisture values are dependent on their initial values and lower than tower measured values and due
to the spatial variation in measured values, the correlation coefficient is also low, and almost the same as ours at 0.47.

LeMone et al. (2010a) simulated fair weather CBL days during the 2002 International H2O Project (IHOP 2002) field campaign in Kansas, USA using the WRF-ARW version 2.1.2, coupled to the NOAH MP land surface scheme. They noted realistically representing surface vegetation, with giving due importance to contributions from the sub-grid vegetation heterogeneity is important in modulating soil moisture, surface flux partitioning and how those would impact CBL growth and
horizontal variation of surface fluxes. In LeMone et al. (2010b) they investigated coherent mesoscale structures in the ABL and underscored the importance of non-periodic boundary conditions and a coupled land surface scheme, to simulate realistic mesoscale circulations and their diurnal evolution as observed during the field experiment days.

Shao et al. (2013) performed 12 hour daytime LES of Selhausen–Merken experimental site in Germany, with multiple land use classes. They had a multi layer canopy model, with 2 m vertical resolution and grid stretching above 80 m, coupled to a
land surface and multi-layered soil model. The simulations were initialised at 08:00 UTC, with radiosonde measured $\theta$, $q$ and wind profiles (mean boundary layer wind $\approx 3.6$ ms$^{-1}$), with periodic boundary conditions. In their sensitivity experiments



they found that the surface heat fluxes simulated by the multi layer canopy model coupled to the land surface model was closer to tower measurements, with the bulk canopy scheme underestimating the near surface $H$ by as much as $100 \ \mathrm{Wm}^{-2}$ in the afternoon.

Heinze et al. (2017) presented an evaluation of LES covering Germany, using the the ICON large eddy model at horizontal resolution of 156 m. The simulations were driven by mesoscale boundary conditions from the Consortium of Small-scale Modelling (COSMO) model at 2.8 km resolution and one-way nested from 625 m to 156 m for several days. They also followed a similar study design as ours, where the LES setup and dataset is intended to serve as a simulation of sub-grid variability for improving and understanding mesoscale models. Their LES results were able to represent the expected temporal
and spatial development of turbulence in comparison with observations from the High Definition Clouds and Precipitation for advancing Climate Prediction (HD(CP)2) project. The also reported substantially larger than observed sensible heat flux in their simulations in combinations with larger net radiation flux as was seen in our study too. The biases they found between simulated and observed ABL profiles were also comparable to the biases in their forcing profiles from the COSMO data, alike the mismatches we saw comparing our simulated profiles with radisonde measured data.

These studies indicate that having the boundary and initial conditions reflect the large scale and observed variability can guide the LES simulated ABL mean state and diurnal variability to be closer to the actually field-measured state. This was also the basis for the implementation of the mesoscale nesting interface in PALM, detailed in Kadasch et al. (2021). Resler et al. (2021) evaluated the PALM model system, over a realistic urban canopy in Prague, Czech Republic for multiple days in Summer and Winter, including PALM's mesoscale nesting, radiative transfer, land surface modules. Their study also reported
strong sensitivity of the results to the accuracy of initialisation and boundary conditions. Moreover, they also noted that the LES solution is partly reflective of the mesoscale model outputs, which our study also points towards. If a comprehensive observational dataset if available before the start of a realistic LES run, it would be beneficial to tune the mesoscale boundary conditions to be reflective of actually observed field experiment conditions.

## 7    Summary and Conclusions

We performed fully coupled LES of the ABL over a heterogeneous, predominantly forested landscape with multiple scales of land surface variability. The simulations were constrained and initialised by land surface properties measured and collected as part of the CHEESEHEAD19 field campaign. Three one way nested models were setup centred around the US-PFa Ameriflux regional tall tower. The simulations were driven by non periodic boundary conditions setup using the NCEP-HRRRv4 data product averaged over the $49 \times 50 \ \mathrm{km}^2$ Parent domain to simulate the diurnal cycles from 22 August 2019, 00:00 CDT till
23 August 2019, 20:00 CDT and from 24 September 2019, 00:00 CDT till 25 September 2019, 2000 CDT . We presented intercomparisons between simulated and observed vertical profiles and near-surface time series data. The simulated profiles and time series correlated well with the observed $\theta$ and $q$ values, with a Pearson's correlation coefficient of 0.9 for almost all cases (Table 3) indicating that the simulations aligned well with the diurnal ABL evolution in the study domain. The mean simulated values are modulated by the imposed HRRR boundary conditions, most strongly reflected in our study for the near





surface $q$ time series with a RMSE of 5.39 $\mathrm{gkg}^{-1}$ for August simulations and 7.19 $\mathrm{gkg}^{-1}$ for the September simulations. Our recommendation with regard to minimising the bias from mesoscale forcing while modelling diurnal evolution of the ABL is similar to that of Resler et al. (2021), to continuously nudge the imposed boundary conditions towards observations or tune the boundary conditions with available observations. Even given this dependence and biases, the LES were able to capture the ABL evolution and characteristics during the intensive observation periods of the CHEESEHEAD19 field campaign.

We believe that our simulation configuration and dataset, in conjunction with the field measurements, can provide a numerical test bed to investigate longstanding micrometeorological research questions concerning unstructured surface heterogeneities such as, corrections for tower measured eddy covariance fluxes, turbulent flux footprints, heterogeneity induced ABL fluxes etc. Moreover, since the model setup was able to realistically represent the observed ABL evolution during the field experiment days, the simulation data can act as a baseline for more in depth explorations on the impacts of sub-grid variability and needed improvements for mesoscale models in a controlled numerical environment.

*Code and data availability.* The PALM model system is freely available from https://palm.muk.uni-hannover.de and is distributed under the GNU General Public Licence v3 (https://www.gnu.org/licenses/gpl-3.0.html). The model source code version 6.0 in revision 21.10-rc.2 used in this article is also available from https://doi.org/10.5281/zenodo.8173281 (Paleri et al., 2023b). The model configurations and inputs of for all simulated cases are available from https://doi.org/10.5281/zenodo.8179066 (Paleri et al., 2023c). All of the CHEESEHEAD19 observations are archived at the NCAR EOL repository at www.eol.ucar.edu/field_projects/cheesehead. Additional information about the project, including descriptions of the sites, photographs, and data plots can be found on the CHEESEHEAD19 website, located at www.cheesehead19.org.



## Appendix A: Adjustment Zone Lengths and Ensemble Convergence



**Figure A1.** Horizontal profiles of 30-minute time-averaged turbulent kinetic energy at 11:00 (a), 13:00 (b) and 15:00 CDT (c) for the 24 September simulations. Horizontal profiles are shown for $0.1z_i$ (solid line), $0.5z_i$ (dashed and dotted) and $0.75z_i$ (dashed). The x-axis shows distance to the inflow boundary. The turbulent kinetic energy was computed at each x,y grid point. Grid points with a similar distance to the inflow boundary were sorted into 1000-equally spaced bins and averaged.





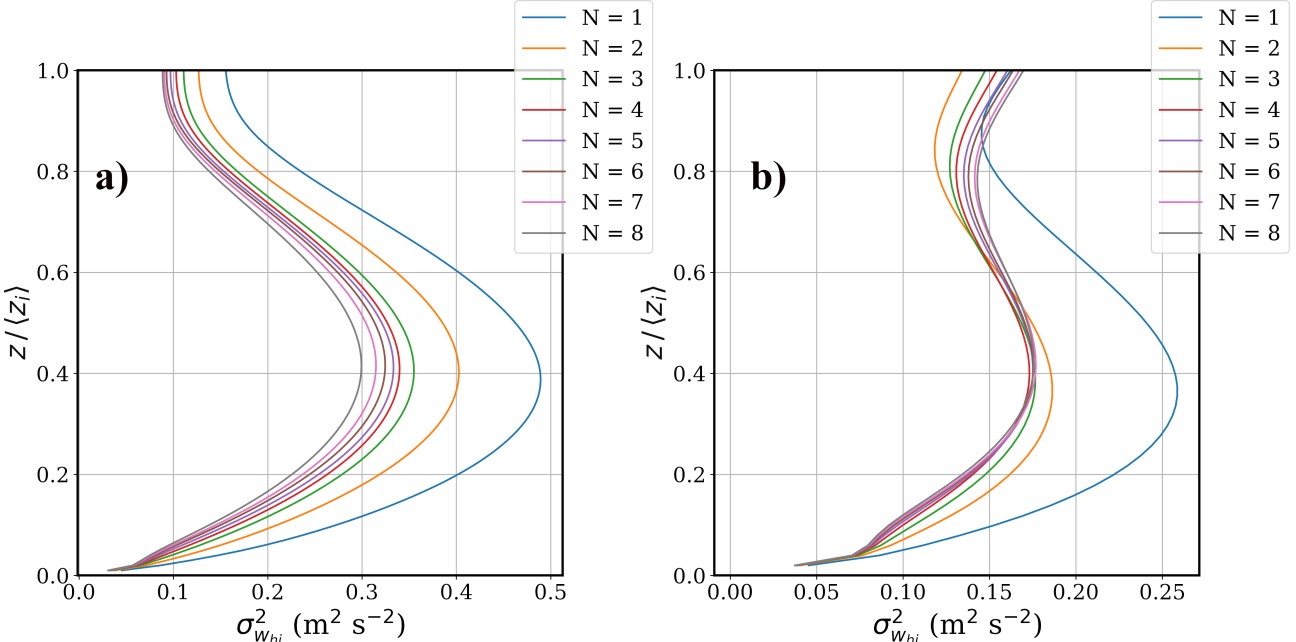

**Figure A2.** Spatial variance of heterogeneity induced vertical wind for increasing number of ensemble members (N), at 12:00 CDT for 23 August (a) and 24 September (b).

*Author contributions.* SP and LW contributed towards modal setup with mentoring and technical support from MS. SP and MS performed
615 initial sensitivity tests. SP, LW, and ARD acquired computational resources. SP performed the final production model runs and the formal
analysis. SP, LW, and ARD were responsible for data curation. SP prepared the original draft. SP, LW, MS, ARD, and MM reviewed and
edited the manuscript drafts. SP and LW created visualisations. ARD, MS and MM provided supervision. ARD and MM acquired funding.

*Competing interests.* The authors declare that they have no conflict of interest.

*Acknowledgements.* This material is based in part upon work supported by the National Science Foundation through the CHEESEHEAD19
620 project (Grant AGS-1822420). The contribution of Luise Wanner was supported by Deutsche Forschungsgemeinschaft (DFG)Award #406980118
and the MICMoR Research School of KIT. For Sreenath Paleri additionall funding was provided by NOAA/Office of Oceanic and Atmo-
spheric Research under NOAA-University of Oklahoma Cooperative Agreement #NA21OAR4320204, U.S. Department of Commerce. We
thank the developers of PALM at the Institute of Meteorology and Climatology of Leibniz Universität Hannover, Germany for the open source
model and their technical support. We would like to acknowledge high-performance computing support from Cheyenne (doi:10.5065/D6RX99HX)
625 provided by NCAR's Computational and Information Systems Laboratory, sponsored by the National Science Foundation.



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
