# Peer review of "Coupled large eddy simulations of land surface heterogeneity effects and diurnal evolution of late summer and early autumn atmospheric boundary layers during the CHEESEHEAD19 field campaign"

_EGUsphere, 2023_

## Referee Comment (RC3)

REVIEW

**Coupled large eddy simulations of land surface heterogeneity effects and diurnal evolution of late summer and early automn atmospheric boundary layers during CHEESEHEAD19 fiels campaign**

S. Paleri, L. Wanner, M. Sühring, A. Desai and M. Mauder

This study presents a LES of 4 IOP days of the CHEESEHEAD19 campaign and evaluates these simultions comparing them with observations. This article is far from publishable as it stands, for a variety of reasons: poorly defined study objectives, poor structure and organisation, numerous contradictions, lack of informations/explanations, lack of justification of the scientific choices and discussions of the results, …

The most important comment I have is that I don't see any results worth publishing. This paper would be a first part about LES evaluation, with a second part about effects of heterogeneity, I would understand. But as it stands, this paper is about an LES whose results are no convincing.

All these factors make the paper difficult to read and understand. I think that the paper has not been sufficiently proofread and worked on by the authors, whose competence I do not doubt. I don't think it's the reviewers' role to list all the flaws in a paper when the co-authors haven't done the work themselves. So I won't make an exhaustive list of the changes needed, but I will illustrate the criticisms with a few examples.

Poorly defined article objectives:
1- The title suggests that the surface heterogeneity effects are simulated. No results in the simulation show the effects of the surface heterogeneity. The authors may have tried to address this point in the very last section (5.1), but the results and analysis are not at all convincing.
2- L92: At the end of the introduction, I understood that the objective of the article is written as follows: "Following through, we ask, can such a LES be used to evaluate mechanisms that generate surface-heterogeneity induced mesoscale circulations in the diurnal ABL? " . No mesoscale circulation induced by surface heterogeneity is shown in these simulations, so the question is not answered in the study.

Article organisation and logic :
1- The field experiment is commented on without introducing Figure 1, which is the only very rough illustration of the experimental set-up.
2- many discussions based on information introduced later in the paper. Some exemple below
  • L104 : CDT introduced L190
  • L105 : Child01 introduced L170
3- Some sections need to be revised to improve the organisation of the ideas. Exemple :
  • section 2.2 starts with the EC towers (L107-110), then continues with the measurements of leaf phenology (L110), comes back over EC tower (L111-113) and ends with Drone base lidar measurement.
  • The introduction to section 4 is another example of a poorly organised paragraph, jumping from one idea to another only to return to the first.
4- The information about LAD profile are in section 3.3.2, normally devoted to Plant canopy Model, whereas too few information about leaf phenology and LAD measurement by drone is given

in section 2.2. What about moving L236 to L252 and associated figures in section 2.2 ?

5-It seems to me that part 3.2 should be presented first, as it is necessary for understanding the following sections.

6- L187 : why the airborne data are mentioned here whereas their are not introduced in section 2 and not used in the study. Also, virtual flight tracks are presented in the section 3.3.2 about Plant Canopy Model (!) and again in section 3.4 about Virtual observational infrastructure… no use since these virtual observations are not used.

7- I don't understand the usefulness of section 3.5. Could be used as an introduction of section 4 ?

8- Figures 12 and 13 could be merged.

Missing information :

An article can refer to previous studies, but it must also be understandable on its own.

1- L98 : before going through the different data used, a rapid general presentation of the experimental set-up would be useful.
- What are the horizontal scales of the surface heterogeneity ?
- What is the surface flux heterogeneity ?
- Are all the EC tower on forest sites ?
- …
  These would be usefull to fully understand the choices made in this LES study.

2- section 3.1 : some methods like « self-nesting » or « offline-nesting » are not at all defined whereas this sentence (L174) : « Employing both the offline-nesting and self-nesting modules lets us include the synoptic scale effects over the simulation domain and model the influence of a heterogeneous land surface and plant canopy over a wide range of scales. » tells that the effects is huge as it can be seen on the model-observation comparisons (section 4.3). These methods are neither defined nor justified.

3- Section 3.3.2 : the Plant Canopy Model is not described : 5 lines over two pages of this section.

Scientific choices justifications :

1- It is written in the abstract that the runs have no cloud simulation ("The simulations were run without clouds which resulted in higher daytime sensible heat fluxes for some scenarios"). No explanation is given on this very important set up choice in the article. L415, the model-observation difference in terms of sensible heat flux is then explained by the lack of clouds in the model. I think the authors should further explain this choice and also justify the interest of realistic simulation in which the clouds are not represented.

2- Section 3.3.2 & fig 2 : A leaf fall is defined for standard forest and wetland forest. Besides it is the first time in the paper that forest over wetland is discussed (and we don't know the proportion), it seems that the leaf fall for standard and wetland are the same. The mean curves are different because the statistic over wetland is really poor. So why the authors defined two curves.

3- L295-296 : Why the authors use different data output frequency for August (30 minutes) and September (15 minutes) IOPS?

4- Section 5: very little is written to justify and explain why the authors want to compare the August and September IOPs in terms of stability and horizontal evolution. What do we learn from this?

5- The choice to assign forest to short grass to avoid double-accounting for surface radiative effects remains a mystery even if I do not doubt that this choice is the good one. This goes with the poor presentation of the PCM.

Insufficient explanations:

1- Line 263-264 : "This helps us to include effects of the spatially heterogeneous plant canopy and high clouds on the simulated surface radiation and flux budgets.". I don't understand why the use of HRRR data over the Parent model domain helps to include effects of the

spatially heteregeneous plant canopy ? It include horizontally heterogeneous surface energy balance between shaded and unshaded surfaces, but what is the link with the heterogeneous plant canopy?

2- L309-312: "Gehrke et al. (2021) discusses this issue, suggesting the role of the SGS model and radiation scheme in combination with the grid resolution as well as the role of the LSM's surface energy balance parameterisation in combination with Monin-Obukhov Similarity Theory based computation of atmospheric fluxes at the first model grid point.". Concerning the important under-estimation of the simulated temperature close to surface in the morning, the explanation given by Gehrke cited by the authors does not help a lot, since all the possible causes are listed there.

3- Concerning the bias between simulated and observed flux, nothing is said about the surface energy balance non closure in the measurements which could, if it was considered here, reduce the difference.

Contradictions:

1- L285: "In this manuscript we focus on 3D data from the Child01 model for 23 August and 24 September simulations, when the model domain encompassed the whole of CBL". In section 4.3, data from 22-23 August and 8-9 September from Child02 are analysed. A discussion would be useful on the effect of ABL developed vertically and encompassed or not in the child01 domain.